# Complete mitogenome of endemic plum-headed parakeet *Psittacula cyanocephala* – characterization and phylogenetic analysis

Prateek Dey[1☯], Sanjeev Kumar Sharma[1☯], Indrani Sarkar[1], Swapna Devi Ray[1], Padmanabhan Pramod[1], Venkata Hanumat Sastry Kochiganti[2], Goldin Quadros[3], Saurabh Singh Rathore[4], Vikram Singh[5], Ram Pratap Singh[1,6]*

1 National Avian Forensic Laboratory, Sálim Ali Centre for Ornithology and Natural History, Coimbatore, Tamil Nadu, India, 2 National Institute of Animal Nutrition and Physiology, Bengaluru, India, 3 Wetland Ecology Division, Sálim Ali Centre for Ornithology and Natural History, Coimbatore, Tamil Nadu, India, 4 Mahatma Gandhi Central University, Motihari, India, 5 Central University of Himachal Pradesh, Dharamshala, India, 6 Department of Life Science, Central University of South Bihar, Gaya, India

☯ These authors contributed equally to this work.
* rampratap@cusb.ac.in

**Data Availability Statement:** All other sequencing and assembly data will be freely available after acceptance. For now, we provide the GenBankAccession No. MT433093.

## Abstract

*Psittacula cyanocephala* is an endemic parakeet from the Indian sub-continent that is wide-spread in the illegal bird trade. Previous studies on *Psittacula* parakeets have highlighted taxonomic ambiguities, warranting studies to resolve the issues. Since the mitochondrial genome provides useful information concerning the species evolution and phylogenetics, we sequenced the complete mitogenome of *P. cyanocephala* using NGS, validated 38.86% of the mitogenome using Sanger Sequencing and compared it with other available whole mitogenomes of *Psittacula*. The complete mitogenome of the species was 16814 bp in length with 54.08% AT composition. *P. cyanocephala mito*genome comprises of 13 protein-coding genes, 2 rRNAs and 22 tRNAs. *P. cyanocephala* mitogenome organization was consistent with other *Psittacula* mitogenomes. Comparative codon usage analysis indicated the role of natural selection on *Psittacula* mitogenomes. Strong purifying selection pressure was observed maximum on *nad1* and *nad4l* genes. The mitochondrial control region of all *Psittacula* species displayed the ancestral avian CR gene order. Phylogenetic analyses revealed the *Psittacula* genus as paraphyletic nature, containing at least 4 groups of species within the same genus, suggesting its taxonomic reconsideration. Our results provide useful information for developing forensic tests to control the illegal trade of the species and scientific basis for phylogenetic revision of the genus *Psittacula*.

## Introduction

*Psittaciformes* is a highly speciose order of class Aves containing 86 genera and 362 distinct species [1]. These intriguing birds are gifted with highly expanded brains [2], superior cognitive ability [3] and vocal communication skills [4]. These birds have been kept as pets since

**Funding:** We thank the Ministry of Environment, Forest and Climate Change, Govt. of India for financial support. RPS and PP acquired the funding.

**Competing interests:** The authors have declared that no competing interests exist.

ancient times because of their beauty, charm, long life span and astonishing ability to imitate many sounds, including human speech [5]. Unfortunately, this has led to widespread increase in parrot poaching and organized illegal trade of these species the world over, making them the most threatened bird species in the world [6, 7].

Plum-headed parakeet *Psittacula cyanocephala*, belonging to the genus *Psittacula* (consisting of 16 long tailed parakeet species of which 13 are extant and 3 extinct) [8], is endemic to Indian sub-continent [9]. This parakeet belongs to one of those bird species which have been widely impacted by illegal live bird trade [10]. Though systematic population assessment of *P. cyanocephala* has never been conducted, it is suspected that, its population is dwindling which can be attributed to ongoing poaching for its illegal trade and habitat destruction [11]. Besides, it is documented that the rapidly changing climate does contribute to the loss of endemic species [12]. Hence, the management and conservation of endemic species such as *P. cyanocephala* would provide an umbrella of protection for various species assemblages within its endemic range [13]. However, to adopt and devise a healthy conservation and management plan, we require complete/ more information regarding behaviour, genetic, physiological, and ecological understanding of *P. cyanocephala*.

*P. cyanocephala* bears close morphological resemblance to *Psittacula roseata* in terms of distinctive head plumage and genetic proximity [8, 14]. A number of studies on evolutionary, morphological, and systematic aspects of the *Psittacula* genus have highlighted taxonomic ambiguities such as the paraphyletic nature of certain species and cladding of the genus *Tanygnathus* within the genus *Psittacula* [8, 14–16]. Recently, Braun and coworkers reconstructed the Maximum Likelihood (ML) based phylogenetic tree of *Psittacula* and other closely related species using a HKY model of two genetic markers (mitochondrial *cob* and nuclear *rag1* gene) and observed that the Asian genus *Psittacula* is paraphyletic [8]. To create monophyletic genera, Braun and coworkers proposed the recognition of genus *Himalayapsitta* Braun, 2016 for *P. himalayana*, *P. finschii*, *P. roseata*, and *P. cyanocephala* [8]. However, the suggested taxonomic revision needs more confirmation using complete mitochondrial genome data.

Mitochondrial genomes are extremely useful in phylogeny and population research of avian taxa because of their inherent properties like small genome size, very less or absence of recombination frequency, maternal inheritance along with highly conserved gene content and evolutionary rate [17–19]. Also, mitochondrial genomes are comparatively more conserved than nuclear genomes during transition events in the evolutionary context, especially in birds [20]. Hence, such unique architecture, organization as well as evolutionary behavior render mitochondrial genomes the ability to carry phylogenetic information more consistently in comparison to nuclear markers [20]. Birds, however, are particularly noteworthy because their mitochondrial genomes are characterized by a gene order different from that in the majority of vertebrate mitogenomes due to rearrangement near the control region [21]. Although, the mitochondrial genome evolves about 10 times faster than the single-copy fraction of the nuclear genome [22], mutations in mitochondrial DNA are largely point mutations with only a few insertions or deletions. Furthermore, it has been reported that complete mitogenomes retain more information than a single gene regarding the evolutionary history of a taxon and provide consistent results compared to nuclear genes [20, 23]. This reduces the effect of homoplasy and frequent stochastic errors in phylogenetic studies [23]. Considering the usefulness of complete mitogenome in phylogenetic and evolutionary studies, we attempted to decode the complete mitogenome of *P. cyanocephala* using Next Generation Sequencing (NGS). Furthermore, we validated NGS data using Sanger sequencing, which is considered as a gold standard for DNA sequencing, for select protein-coding genes to rule out any possible sequencing error.

We successfully decoded the complete mitochondrial genome sequence of *P. cyanocephala* for the first time. The newly sequenced mitogenome was validated and compared with other

available mitogenomes of related genera to 1) know structural and genomic features of *P. cyanocephala* mitogenome and its relatedness with other available *Psittacula* mitogenomes, 2) identify codon usage and selection pressure on the protein-coding genes (PCGs) using bioinformatics and evolutionary analysis across the numerous parrot genera, and 3) build and elucidate comprehensive phylogeny relationships using all the available mitogenomes including that of *P. cyanocephala*.

## Materials and methods

### Sample collection and DNA extraction

Sample of *Psittacula cyanocephala* blood (~50 μl) was obtained from the Veterinary Transit Treatment Centre (21˚10'10"N, 79˚03'30"E), Nagpur, Maharashtra with due permission from Forest Department of Maharashtra [Desk-22(8)/Research/CR-8(19–20)/769/2019-2020] and used for this study. The blood sample was stored in Queen's lysis buffer [24] and transported under suitable conditions to National Avian Forensic Laboratory (NAFL) at SACON, Coimbatore, Tamil Nadu. About 40μl of blood was digested in lysis buffer (10 mMTris, 10 mM EDTA, 10% SDS and 40μg Proteinase K) and subsequently used for DNA extraction using the Phenol-Chloroform-Isoamyl Alcohol method with minor modifications [25]. The extracted DNA was quantified using DeNovix Spectrophotometer (DeNovix Inc., Delaware, USA), Qubit-4 Fluorometer (ThermoFisher Scientific, USA) and subsequently stored at -80˚C till used. The specimen DNA was deposited in the Avian Biobank at NAFL under the accession code NAFL/0305/DNA/180220.

### Library preparation

For library preparation, 1.1 micrograms of extracted DNA was utilized as starting material for TruSeq DNA PCR-Free library preparation kit (Illumina Inc., USA). The DNA was fragmented using focused ultrasonicator (Covaris M220, USA) to the desired length of 350 base pairs as per the protocol recommended by the manufacturer. The size of the fragmented DNA population was checked using Fragment Analyzer (Agilent, USA) and made sure the majority of fragment sizes were in the desired range. Following TruSeq DNA PCR-Free library preparation kit protocol, clean-up of the fragmented DNA was performed using magnetic beads provided in the kit. The process of fragmentation creates overhangs at 3' and 5' ends of the input DNA. These ends were converted into blunt ends using the kit protocol. After the end repair, the appropriate library size was selected using different ratios of sample purification beads provided in the kit. The 3' end of the blunt fragments was adenylated with a single "A" nucleotide to prevent them from ligating to each other. Adapter ligation was then carried out following the kit protocol. After completion of library preparation, the mean peak size of the library prepared was checked using Fragment Analyzer. Quantification of the prepared library was carried out using QIAseq Library Quant Assay Kit (Qiagen N.V., Germany). The library was sequenced using NextSeq550 (Illumina Inc., USA) and at the end of the sequencing run, high-quality paired-end reads were obtained.

### Mitogenome assemblage

FastQC [http://www.bioinformatics.babraham.ac.uk/projects/fastqc/] was used to check the quality of the sequence data. At first, the *de-novo* assembly of the genome was attempted using SPAdes [26]. However, due to low coverage (~ 4.7X) of the whole genome of *P. cyanocephala*, successful retrieval of complete de-novo mitogenome sequence wasn't possible. Hence the reads were corrected using SPAdes and then mapped directly onto the reference genome

(*Psittacula roseata* (NC045379.1)) using Bowtie2 [27]. A very high coverage (~ 90X) of the mitogenome was obtained using this approach. The final alignment was visualized using IGV software [28].

## PCR amplification and sequencing of the protein coding genes using Sanger sequencing

Select protein-coding genes (*cox1*, *cox2*, *atp8*, *atp6*, *nad1*, *nad2*, *nd3 and cob*) of *P. cyanocephala* mitogenome were amplified to verify the mitogenome generated using NGS data. The genes were amplified using published primers (S1 and S2 Tables) designed by our lab [29]. Desired amplicons were amplified using thermocycler as per the S1 and S2 Tables. The PCR amplified amplicons were gel purified, and cycle sequenced with Big Dye Terminator ver. 3.1 (Applied Biosystems, Foster City, CA) using Applied Biosystems Genetic Analyzer 3500 (Applied Biosystems, Foster City, CA). Difficulty was faced during cycle sequencing of certain templates due to possible secondary structure formation in respective PCR amplicons. Various PCR adjuvants such as betaine and formamide in suitable concentrations and volumes were used to sequence the templates successfully.

## Gene annotation

The complete mitogenome of *P. cyanocephala* was annotated manually. Using MitosWeb-server all the PCGs, rRNAs and tRNAs were predicted and used as a template for annotation of the newly assembled mitogenome [30, 31]. The tRNA positions and secondary structures were predicted using tRNAscan-SE2.0 [32], verified with the MITOSWeb-server results and carefully annotated on the newly sequenced *P. cyanocephala* mitogenome. The rRNA genes were initially predicted by MitosWeb-server and the boundaries were confirmed by aligning the *P. cyanocephala* mitogenome with other *Psittacula* mitogenomes [*P. alexandri* (NC045378.1) [33], *P. derbiana*(NC042409.1) [34], *P. eupatria*(NC042765.1) [34], *P. krameri* (MN065674.1) [35] and *P. roseata* (NC045379.1)] from GenBank (Table 1). Protein coding regions (open reading frames (ORF)) were identified in the sequenced *P. cyanocephala* mitogenome by using NCBI ORF FINDER [36]. The identified regions were translated using ExPASy-Translate Tool [37] with settings for the vertebrate mitochondrial genetic code. ExPASy-Translate Tool was used to check each ORF region for a protein-coding gene (PCG) with correct start and stop codons The formula "AT-skew = (A-T)/(A+T)" and "GC-skew = (G-C)/(G+C)" was used to calculate nucleotide composition skew [38]. A circular genome map was generated with the help of the CGView Server and edited manually [39].

## Codon usage and evolutionary analysis

Complete mitochondrial genomes of members from *Agapornis*, *Amazona*, *Ara*, *Eupsittula*, *Brotogeris*, *Cacatua*, *Calyptorhynchus*, *Coracopsis*, *Eclectus*, *Eolophus*, *Forpus*, *Guaruba*, *Lorius*, *Melopsittacus*, *Nestor*, *Poicephalus*, *Primolius*, *Prioniturus*, *Probosciger*, *Psephotellus*, *Psittacula*, *Psittacus*, *Psittrichas*, *Pyrrhura*, *Rhynchopsitta*, *Strigops*, *Tanygnathus* and *Trichoglossus* genus were downloaded from NCBI GenBank database (Table 1). These genomes were analyzed along with the newly sequenced mitogenome of *P. cyanocephala*.

CodonW was used for the detailed codon usage analysis [40]. This analysis includes the percentage of AT and GC, nucleotide bias at the third position of codons, frequency of optimal codons (Fop), Effective number of codons (ENc) and Relative Synonymous Codon Usage (RSCU). Overall codon and amino acid usage of the newly sequenced mitogenome was represented by roseplots. Comparative codon and amino acid usage among 45 select mitogenomes were represented by heatmaps. Roseplots and heatmaps were generated using R software [41].

**Table 1. List of the 44 *Psittaciformes* species used for comparative analysis in this study with their GenBank accession numbers.**

| Family | Genus | Species | GenBank No. |
|---|---|---|---|
| Cacatuidae | *Cacatua* | *Cacatua moluccensis* | MH133972.1 |
| Cacatuidae | *Cacatua* | *Cacatua pastinator* | NC040142.1 |
| Cacatuidae | *Calyptorhynchus* | *Calyptorhynchus baudinii* | MH133969.1 |
| Cacatuidae | *Calyptorhynchus* | *Calyptorhynchus lathami* | JF414241.1 |
| Cacatuidae | *Calyptorhynchus* | *Calyptorhynchus latirostris* | JF414243.1 |
| Cacatuidae | *Eolophus* | *Eolophus roseicapilla* | NC040154.1 |
| Cacatuidae | *Probosciger* | *Probosciger aterrimus* | MH133970.1 |
| Nestoridae | *Nestor* | *Nestor notabilis* | KM611472.1 |
| Psittacidae | *Aratinga* | *Aratinga acuticaudata* | JQ782214.1 |
| Psittacidae | *Amazona* | *Amazona aestiva* | KT361659.1 |
| Psittacidae | *Amazona* | *Amazona barbadensis* | JX524615.1 |
| Psittacidae | *Amazona* | *Amazona ochrocephala* | KM611467.1 |
| Psittacidae | *Ara* | *Ara militaris* | KM611466.1 |
| Psittacidae | *Ara* | *Ara severus* | KF946546.1 |
| Psittacidae | *Aratinga* | *Aratinga pertinax* | HM640208.1 |
| Psittacidae | *Brotogeris* | *Brotogeris cyanoptera* | HM627323.1 |
| Psittacidae | *Forpus* | *Forpus modestus* | HM755882.1 |
| Psittacidae | *Forpus* | *Forpus passerinus* | KM611470.1 |
| Psittacidae | *Guaruba* | *Guaruba guarouba* | NC026031.1 |
| Psittacidae | *Poicephalus* | *Poicephalus gulielmi* | MF977813.1 |
| Psittacidae | *Primolius* | *Primolius couloni* | KF836419.1 |
| Psittacidae | *Primolius* | *Primolius maracana* | KJ562357.1 |
| Psittacidae | *Psittacus* | *Psittacus erithacus* | KM611474.1 |
| Psittacidae | *Pyrrhura* | *Pyrrhura rupicola* | KF751801.1 |
| Psittacidae | *Rhynchopsitta* | *Rhynchopsitta terrisi* | KF010318.1 |
| Psittaculidae | *Agapornis* | *Agapornis lilianae* | NC045369.1 |
| Psittaculidae | *Agapornis* | *Agapornis nigrigenis* | NC045367.1 |
| Psittaculidae | *Agapornis* | *Agapornis pullarius* | NC045368.1 |
| Psittaculidae | *Agapornis* | *Agapornis roseicollis* | EU410486.1 |
| Psittaculidae | *Eclectus* | *Eclectus roratus* | KM611469.1 |
| Psittaculidae | *Lorius* | *Lorius chlorocercus* | MN515396.1 |
| Psittaculidae | *Melopsittacus* | *Melopsittacus undulatus* | EF450826.1 |
| Psittaculidae | *Prioniturus* | *Prioniturus luconensis* | KM611473.1 |
| Psittaculidae | *Psephotellus* | *Psephotellus pulcherrimus* | KU158195.1 |
| Psittaculidae | *Psittacula* | *Psittacula alexandri* | NC045378.1 |
| Psittaculidae | *Psittacula* | *Psittacula derbiana* | NC042409.1 |
| Psittaculidae | *Psittacula* | *Psittacula eupatria* | NC042765.1 |
| Psittaculidae | *Psittacula* | *Psittacula krameri* | MN065674.1 |
| Psittaculidae | *Psittacula* | *Psittacula roseata* | NC045379.1 |
| Psittaculidae | *Tanygnathus* | *Tanygnathus lucionensis* | KM611480.1 |
| Psittaculidae | *Trichoglossus* | *Trichoglossus rubritorquis* | MN182499.1 |
| Psittrichasiidae | *Coracopsis* | *Coracopsis vasa* | KM611468.1 |
| Psittrichasiidae | *Psittrichas* | *Psittrichas fulgidus* | KM611475.1 |
| Strigopidae | *Strigops* | *Strigops habroptilus* | AY309456.1 |

Evolutionary constraints on the individual protein-coding genes in terms of selection pressure were estimated by the ratio of non-synonymous substitution to the synonymous

substitution rate (dN/dS). Maximum likelihood approach was adopted for this analysis, where dN/dS<1 indicates neutral or purifying selection and dN/dS>1 indicates positive/ Darwinian selection/ mutational pressure [42]. *PAL2NAL* program embedded in Phylogenetic Analysis by Maximum Likelihood (PAML) package was used for this analysis. We exploited the stand-alone Linux version of this server [42]. The protein-coding genes of *P. cyanocephala* were assessed against the protein-coding genes of other investigated genomes as mentioned earlier. The codons used in mRNAs were aligned and were subjected to dN/dS analysis. The orthologous pair of gene sequences was given as input file and the dN/dS value for that gene was calculated.

### Genetic divergence and phylogenetic analysis

Divergence analysis is one of the most popular ways to estimate the cumulative differences among closely related members of a species that separated geographically leading to speciation [43] (allopatric or peripatric speciation). Pairwise base substitutions per site can provide an idea about the genomic distance and evolutionary history amongst the investigated genomes. Both species-level and genus-level divergence analysis was performed. For species-level analysis, each genome was taken as a separate entity while in genus-level analysis members of the same genus were considered as one group. For genus-level analysis members of the same genus were partitioned into a distinct group in MEGA X and average distances between the groups (i.e. generas) was calculated. The species-level analysis elucidates genetic distances at an individual level whereas the genus-level approach helps us to understand the average genetic distances between the grouped taxon from a different point of view. Thus, all considered organisms were clustered into 28 different groups which corresponds to 28 genera. The divergence among these groups was calculated using the Jukes-cantor model in MEGA X software [44].

For the construction of the phylogenetic tree, all the selected sequences were concatenated to 13 PCGs. Species across 28 genera of parrots and parakeets were used for tree construction. Emphasis was given to genus *Psittacula*, *Tanygnathus* and *Eclectus* during phylogenetic analysis. In total, 45 mitogenomes were concatenated to their 13 PCGs and aligned using the MUSCLE alignment program. Phylogenetic tree analysis was carried out by Maximum likelihood (ML) and Bayesian inferences (BI) algorithms following two strategies. At the first instance, a best-fit nucleotide substitution model of TVM+F+G4 was identified for our data set using the Akaike information criterion (AIC) and Bayesian information criterion (BIC) in ModelFinder [45]. ML tree was constructed through IQ-TREE [46] version 1.6.12 using the TVM+F+G4 model with 10,000 bootstrap replicates. For BI analysis, GTR+I+G model was selected from BIC scores in ModelFinder by following previous studies [30]. Mr.Bayes [47] software was used to perform BI analysis following four independent chains running for 100000 generations, sub-sampling every 1000 generations and using a burn-in of 100 generations. FigTree [48] ver1.4.4 was used to edit the resulting phylogenetic trees. The convergence and mixing of Bayesian Markov chains was assessed by calculating the average standard deviation of split frequencies along with effective sample size of the trace respectively.

## Results and discussion

### Mitogenome organization and gene arrangement

The complete mitochondrial genome of *P. cyanocephala* was assembled and submitted to Gen-Bank (Accession No. MT433093). Sanger sequencing of eight PCGs (*cox1*, *cox2*, *atp8*, *atp6*, *nad1*, *nad2*, *nd3* and *cob*) showed 99–100% similarity with NGS data. We considered sequence similarity of more than 99% as robust enough for sequence verification in this study. A total of

6595 base pairs comprising the eight PCGs were Sanger sequenced, which makes 38.86% of the total *P. cyanocephala* mitogenome. The mitogenome was 16,814 base pairs long, which is in agreement with the average size of the complete mitogenomes (16827 bp, Table 1) of other *Psittacula* species. The mitogenome of *P. cyanocephala* is composed of 13 protein-coding genes (PCGs), 22 transfer RNAs (tRNA), 2 ribosomal RNA (rRNA) and a mitochondria control region (or D-loop) (Table 2) reported earlier in other birds [1, 30, 49, 50, 52]. Majority of the genes including 12 PCGs, 14 tRNA, 2 rRNAs and the D-loop were located on the heavy (H or +)

**Table 2. Summary of *Psittacula cyanocephala* mitogenome.**

| Gene | Start | Stop | Strand | Length | Intragenic Nucleotides | Anti-Codon | Start Codon | Stop codon |
|---|---|---|---|---|---|---|---|---|
| tRNA^Phe | 1 | 64 | + | 64 | 0 | GAA | | |
| 12s rRNA | 65 | 1040 | + | 976 | 0 | | | |
| tRNA^Val | 1041 | 1112 | + | 72 | 1 | TAC | | |
| 16s rRNA | 1114 | 2697 | + | 1584 | 0 | | | |
| tRNA^Leu | 2698 | 2771 | + | 74 | 5 | TAA | | |
| NAD1 | 2777 | 3757 | + | 981 | -2 | | ATG | TAA |
| tRNA^Ile | 3756 | 3826 | + | 71 | 5 | GAT | | |
| tRNA^Gln | 3832 | 3902 | _ | 71 | 0 | TTG | | |
| tRNA^Met | 3903 | 3970 | + | 68 | 0 | CAT | | |
| NAD2 | 3971 | 5010 | + | 1040 | 0 | | ATG | TA(A) |
| tRNA^Trp | 5011 | 5081 | + | 71 | 1 | TCA | | |
| tRNA^Ala | 5083 | 5150 | _ | 68 | 1 | TGC | | |
| tRNA^Asn | 5152 | 5227 | _ | 76 | 2 | GTT | | |
| tRNA^Cys | 5230 | 5296 | _ | 67 | 0 | GCA | | |
| tRNA^Tyr | 5297 | 5366 | _ | 70 | 9 | GTA | | |
| COI | 5376 | 6923 | + | 1548 | 0 | | GTG | AGG |
| tRNA^Ser | 6924 | 6989 | _ | 66 | 4 | TGA | | |
| tRNA^Asp | 6994 | 7062 | + | 69 | 2 | GTC | | |
| COII | 7065 | 7748 | + | 684 | 1 | | ATG | TAA |
| tRNA^Lys | 7750 | 7818 | + | 69 | 1 | TTT | | |
| Atp8 | 7820 | 7987 | + | 168 | -10 | | ATG | TAA |
| Atp6 | 7978 | 8661 | + | 684 | -1 | | ATG | TAA |
| COIII | 8661 | 9444 | + | 784 | 0 | | ATG | T(AA) |
| tRNA^Gly | 9445 | 9513 | + | 69 | 0 | TCC | | |
| NAD3 | 9514….9686 | 173 | + | | | | ATA | TA(A) |
| | 9688….9864 | 177 | | | 0 | | | |
| tRNA^Arg | 9865 | 9933 | + | 69 | 1 | TCG | | |
| NAD4L | 9935 | 10231 | + | 297 | -7 | | ATG | TAA |
| NAD4 | 10225 | 11617 | + | 1393 | 0 | | ATG | T(AA) |
| tRNA^His | 11618 | 11686 | + | 69 | 0 | GTG | | |
| tRNA^Ser | 11687 | 11752 | + | 66 | 0 | GCT | | |
| tRNA^Leu | 11753 | 11822 | + | 70 | 0 | TAG | | |
| NAD5 | 11823 | 13634 | + | 1812 | 10 | | ATG | TAA |
| CytB | 13645 | 14784 | + | 1140 | 0 | | ATG | TAA |
| tRNA^Thr | 14785 | 14853 | + | 69 | 3 | TGT | | |
| tRNA^Pro | 14857 | 14925 | _ | 69 | 1 | TGG | | |
| NAD6 | 14927 | 15445 | _ | 519 | 1 | | ATG | TAG |
| tRNA^Glu | 15447 | 15497 | _ | 51 | 0 | TTC | | |
| D-loop | 15498 | 16814 | _ | 1317 | | | | |

strand, whereas 8 tRNAs (*trn^Ala^*, *trn^Cys^*, *trn^Glu^*, *trn^Asn^*, *trn^Pro^*, *trn^Gln^*, *trn^Ser^*(AGU) *and trn^Tyr^* and 1 PCG (*nad6*) were located on the light (L or -) strand (Fig 1). Earlier workers have reported similar arrangement of genes in parrots [1, 49, 50] and other birds [30, 51, 52].

The total base composition of *P. cyanocephala* mitogenome was A-5365 (31.9%), T-3728 (22.2%), G-2216 (13.2%) and C-5505 (32.7%). The A+T (54.08%) content was higher than the G+C content (45.92%, Table 3). High A+T content is typical of most vertebrate orders, including the members of *Psittaciformes* order and certain other birds [30, 51, 52]. The overall A-T and G-C skew were 0.1800 and -0.4259, respectively (Table 3), indicating higher A nucleotide than T nucleotide, and higher C nucleotide than G nucleotide, respectively. The A-T and G-C skew values also indicated that *Psittacula* species are having higher A and C content than T and G as reported earlier in certain other birds [30, 51]. The skew values of A-T and G-C revealed in *P. cyanocephala* are nearly identical to what is reported in other species of *Psittacula* genus. The A-T and G-C skew values of *P. alexandri*, *P. derbiana*, *P. eupatria*, *P. krameri* and *P. roseata* were in the range of 0.1588 to 0.1836 and -0.4271 to -0.4151, respectively (Table 4).

Gene arrangement analysis revealed the following pattern in *P. cyanocephala*: *trnF>rrnS>trnV>rrnL>trnL2>nad1>trnI>trnQ>trnM>nad2>trnW>trnA>trnN>trnC>-trnY>cox1>trnS2>trnD>cox2>trnK>atp8>atp6>cox3>trnG>nad3>trnR>nad4l>nad4>-trnH>trnS1>trnL1>nad5>cob>trnT>trnP>nad6>trnE*. The gene arrangement pattern was identical with other *Psittacula* members (S1 Fig).

## Protein-coding genes

The total length of all the PCGs of *P. cyanocephala* was 11,400 base pairs constituting about 67.8% of the entire mitogenome. The occurrence and sequence of all the PCGs of *P. cyanocephala* mitogenome are following other avian mitochondrial genomes [30, 51, 52]. The entire length of all the PCGs translated into 3789 amino acid coding codons excluding stop codons. 11 PCGs of the *P. cyanocephala* mitogenome initiated with the start codon ATG, i.e. *nad1*, *nad2*, *cox2*, *atp8*, *atp6*, *cox3*, *nad4l*, *nad4*, *nad5*, *cob*, and *nad6*. Exceptions were *cox1* and *nad3* initiated with a start codon GTG and ATA, respectively. TAA was the most prevalent stop codon being punctuated in sequences of *nad1*, *nad2*, *cox2*, *atp8*, *atp6*, *cox3*, *nad3*, *nad4l*, *nad4*, *nad5* and *cob*. The stop codon TAG was associated with the PCG *nad6*, whereas the *cox1* gene had a stop codon of AGG. The TAA stop codons of *nad2*, *cox3*, *nad3* and *nad4* genes are partially incomplete and were completed by adding of polyadenylated tails at the 3' ends. Overlapping of sequences was observed only in PCGs of *atp8*, *atp6* (10 nucleotides) and *nad4l*, *nad4* (7 nucleotides). Many of the codon features mentioned here found to be similar across various avian orders [53]. However, within the genus *Psittacula*, start codons remained almost the same for respective genes irrespective of the species. ATG was the most frequent start codon of the PCGs in all *Psittacula* species investigated in this study (*P. cyanocephala*, *P. alexandri*, *P. derbiana*, *P. eupatria*, *P. krameri* and *P. roseata*). The exception of *cox1* (start codon GTG) and *nad3* (start codon ATA) genes, also holds across all the *Psittacula* species investigated so far.

In the case of stop codons, TAA was the most prevalent stop codon followed by TAG, with the *cox1* gene punctuated by AGG stop codon across all the *Psittacula* species studied so far. The sequence overlaps of *atp8*, *atp6* and *nad4l*, *nad4* genes is common amongst all the *Psittacula* species analyzed in this study.

Codon usage analysis revealed that either adenine or cytosine was generally at the third positions of the codons. In *P. cyanocephala*, A3 (0.40) dominated over C3 (0.38) (Fig 2A). This was also validated with codon usage-based roseplots (Fig 2B). For other genera, except *Agapornis*, adenine and cytosine were dominating at the third position of codons. *Agapornis* showed

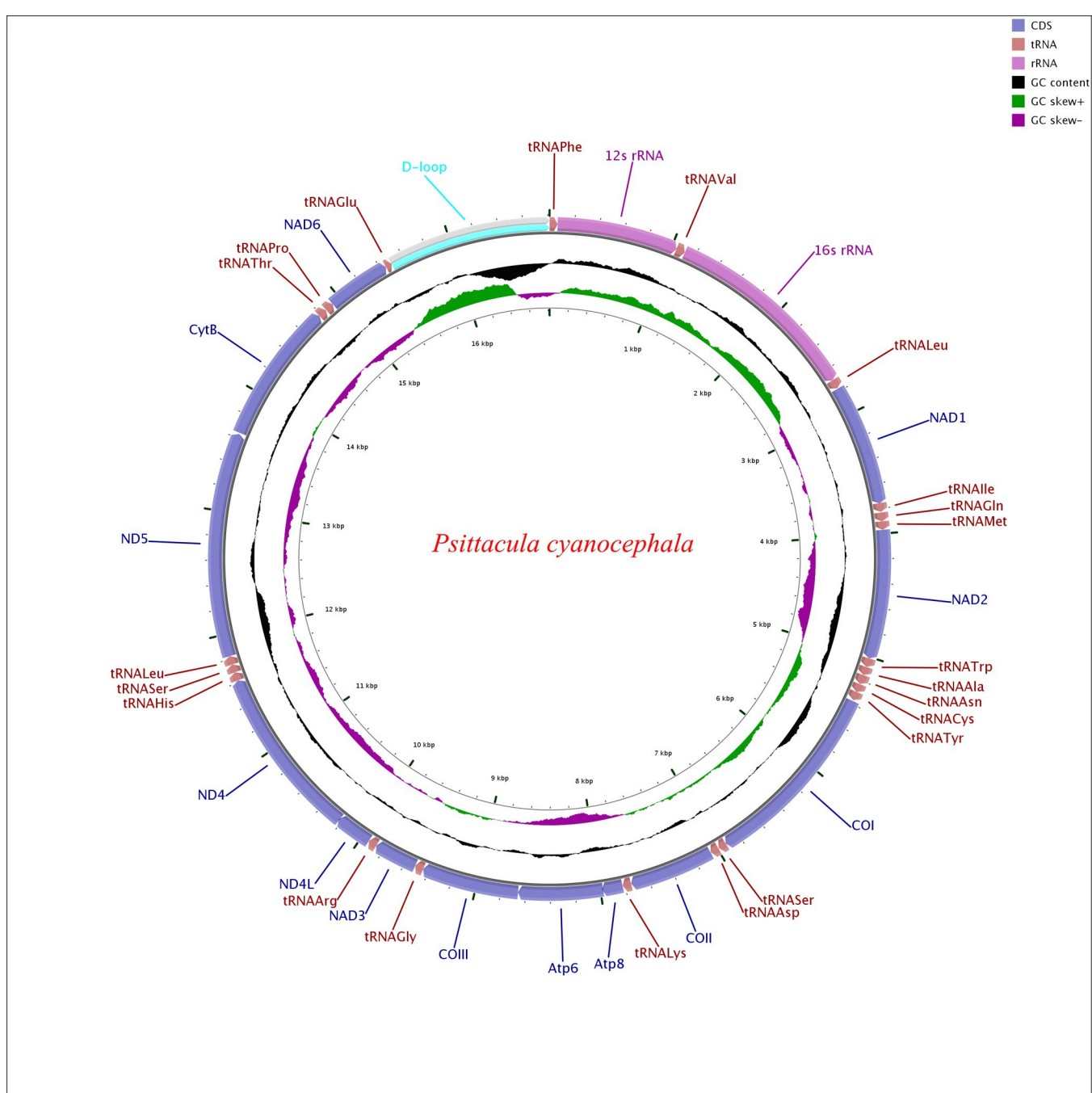

**Fig 1. Circular map of the *Psittacula cyanocephala* mitochondrial genome.** Various genes are represented with different colour blocks. Gene transcription direction is indicated by arrows. Colour codes and legends are displayed at the upper right side of the figure. Black sliding window indicated the GC content of all the regions and GC skew through green and violet colour sliding windows. The figure was drawn by CGView Online server (http://stothard.afns.ualberta.ca/cgview_server/) and edited in PaintDotnet tool.

a C3 bias at codons' third position. The amino acid usage-based heatmap along with roseplot of amino acid of *P. cyanocephala* (Figs 2C and 3B) showed Leucine, Isoleucine, Proline, Serine and Threonine were dominant than other amino acids in the mitogenome of *P. cyanocephala*. The effective number of codons (ENc) using plot analysis (Fig 2E), revealed that all the

**Table 3. Nucleotide composition and skew values for *Psittacula cyanocephala* mitogenome.**

| *P. cyanocephala* | Size(bp) | A% | C% | G% | T% | A+T% | G+C% | AT-skew | GC-skew |
|---|---|---|---|---|---|---|---|---|---|
| mtDNA | 16814 | 31.9 | 32.7 | 13.2 | 22.2 | 54.08 | 45.92 | 0.1800 | -0.4259 |
| PCGs | 11400 | 31.7 | 34.8 | 11.5 | 22.0 | 53.73 | 46.27 | 0.1794 | -0.5025 |
| tRNA | 1486 | 33.8 | 26.0 | 16.8 | 23.4 | 57.27 | 42.73 | 0.1821 | -0.2157 |
| rRNA | 2560 | 34.3 | 30.6 | 17.8 | 17.3 | 51.56 | 48.44 | 0.3287 | -0.2645 |
| D-loop | 1317 | 26.9 | 26.4 | 15.2 | 31.5 | 58.39 | 41.61 | -0.0793 | -0.2700 |

investigated mitogenomes were well below the curve of selection pressure. The ENc plot primarily reflects the mutational bias (Fig 2E). Plotting ENc against the GC3 thus provides an evidence of whether mutational pressure/natural selection is acting on the PCGs of an organism. An umbrella-line is drawn with the "expected ENc value" (value assuming only mutational pressure is acting on the considered genes) and is compared to the observed ENc values. Several previous reports have stated, if the observed ENc values exceeds the expected ENc values it will depict complete mutational pressure on the respective genes [54, 55]. However, if the observed value is less than the expected value it is due to the selection pressure, lowering the effective number of codons. In the Fig 2E, all the plotted points have placed below the umbrella-line indicating the selection pressure over mutational bias on those genes. This clearly indicated that all investigated mitogenomes were translationally efficient, and natural selection was playing a crucial role on these genomes. From RSCU analysis, a set of optimal codons among the studied mitogenomes was identified. These optimal codons were- GCC(A), UGC(C), GAC(D), GAA(E), UUC(F), GGA(G), CAC(H), AUC(I), AAA(K), CUA(L), CUC (L), UUA(L), AUA(M), AAC(N), CCC(P), CCA(P), CAA(Q), CGC(R), AGC(S), UCC(S), ACA(T), ACC(T), GUC(V), GUA(V),UGA(W) and UAC(Y) (Fig 2D and S3 Table). Heat-maps based on codon usage (Fig 3A) showed CUA (L), AUC (I), ACC (T), CUC (L), UUC (F) and ACA (T) to be the most used codons among the select mitogenomes. Amino acid usage heat-map also showed Leucine, Isoleucine and Threonine as the most-used amino acids (Fig 3B).

## Ribosomal and transfer RNA genes

Two ribosomal RNAs were packaged into the mitogenome of *P. cyanocephala* as in most vertebrates. The smaller 12s rRNA gene comprised of 976 base pairs with an A+T content of 50.72% and the larger 16s rRNA gene comprised of 1584 base pairs with an A+T content of 52.08% (Table 3). The rRNAs were nestled between *tRNA^Phe* and *tRNA^Leu*, with *tRNA^Val* separating 12s and 16s RNA. The A+T content in 12s rRNA and 16s rRNA of other *Psittacula* species compared in this study were in the range of 49.8–51.13% and 51.5–52.5% respectively, similar to the nucleotide contents of the *P. cyanocephala* rRNAs. Birds are generally characterized by similar positioning of and high A+T content in their rRNAs, reported across many

**Table 4. Comparative nucleotide composition and skew values for five *Psittacula* species compared in this study.**

| | Size | A% | C% | G% | T% | A+T% | G+C% | AT-skew | GC-skew |
|---|---|---|---|---|---|---|---|---|---|
| *P. roseata* | 16814 | 31.9 | 32.9 | 13.2 | 22.0 | 53.9 | 46.12 | 0.1836 | -0.4271 |
| *P. eupatria* | 17139 | 31.7 | 33.2 | 13.1 | 22.0 | 53.7 | 46.27 | 0.1806 | -0.4344 |
| *P. derbiana* | 16887 | 30.9 | 32.8 | 13.9 | 22.4 | 53.3 | 46.74 | 0.1594 | -0.4043 |
| *P. alexandri* | 16883 | 30.9 | 32.8 | 13.9 | 22.4 | 53.3 | 46.77 | 0.1594 | -0.4041 |
| *P. krameri* | 16413 | 31.0 | 32.9 | 13.6 | 22.5 | 53.5 | 46.49 | 0.1588 | -0.4151 |

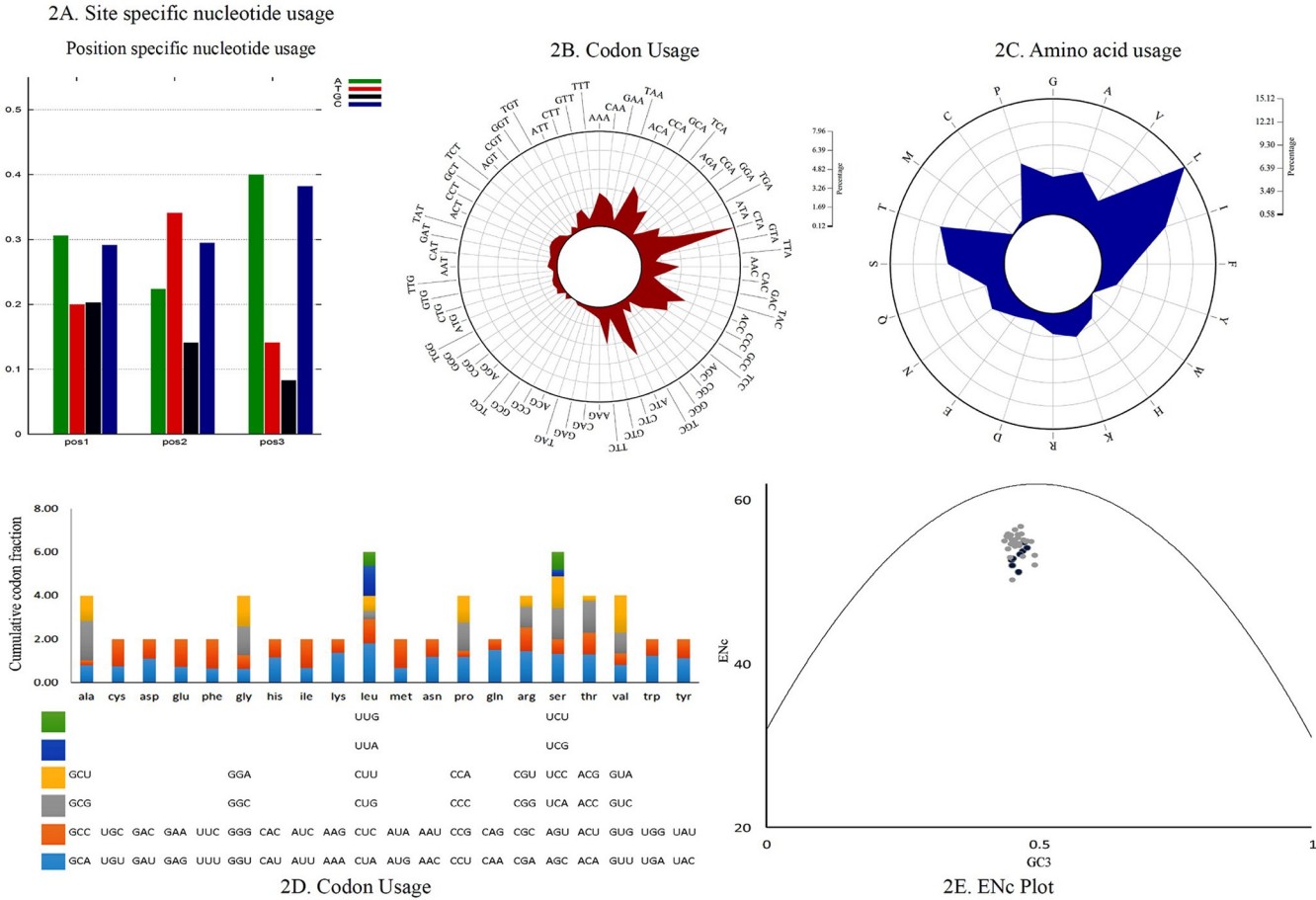

**Fig 2.** (A) Position-specific nucleotide usage in *P. cyanocephala* mitogenome. (B) Roseplot based on codon usage of *P. cyanocephala* mitogenome. (C) Roseplot based on amino acid usage of *P. cyanocephala* mitogenome. (D) RSCU analysis of *P. cyanocephala* mitogenome. X-axis represents the codon families with different colour patches. Cumulative codon fraction is plotted on Y-axis. (E) ENC vs GC3 plot revealed the analyzed mitogenomes are translationally efficient and natural selection was playing a crucial role on their evolution.

avian orders [51, 56]. As such the rRNAs of the *Psittacula* species compared in this study displayed similar positioning and characteristics of rRNAs typical of avian orders.

As in other avian species, 22 tRNAs were identified in the mitogenome of *P. cyanocephala* in this study and their secondary structures were interpreted. The length of the tRNAs varied from 64 base pairs to 74 base pairs (Table 2). The total concatenated length of all tRNAs present in the mitogenome was calculated to be 1486 base pairs. The A-T skew was calculated at 0.1821 and G-C skew at -0.2157, indicating higher A (33.8%) content than T (23.4%) and higher C (26%) content than G (16.8%) (Table 3). The secondary structures of all the tRNAs displayed a cloverleaf model; the exception was *tRNA*$^{Ser}$ (GCU), which was missing the entire dihydrouridine arm (Fig 4). One of the key features of tRNA secondary structure is the presence of wobble base pairing. This can often substitute the Watson-crick base pairing and provide thermodynamic stability to tRNA. An extensive study on the tRNAs is crucial for the proper understanding of functional and structural features of mitogenomes. Highest wobble base pairing was found in *trn*$^{Gln}$ followed by *trn*$^{Pro}$ and *trn*$^{Cys}$. Both *trn*$^{Gln}$ and *trn*$^{Pro}$ had three wobble base pairings each in amino acid acceptor arm, DU stem and anticodon stem. However, it was also found *trn*$^{Gln}$ had an extra wobble base pairing on TψC stem. *trn*$^{Cys}$ had wobble

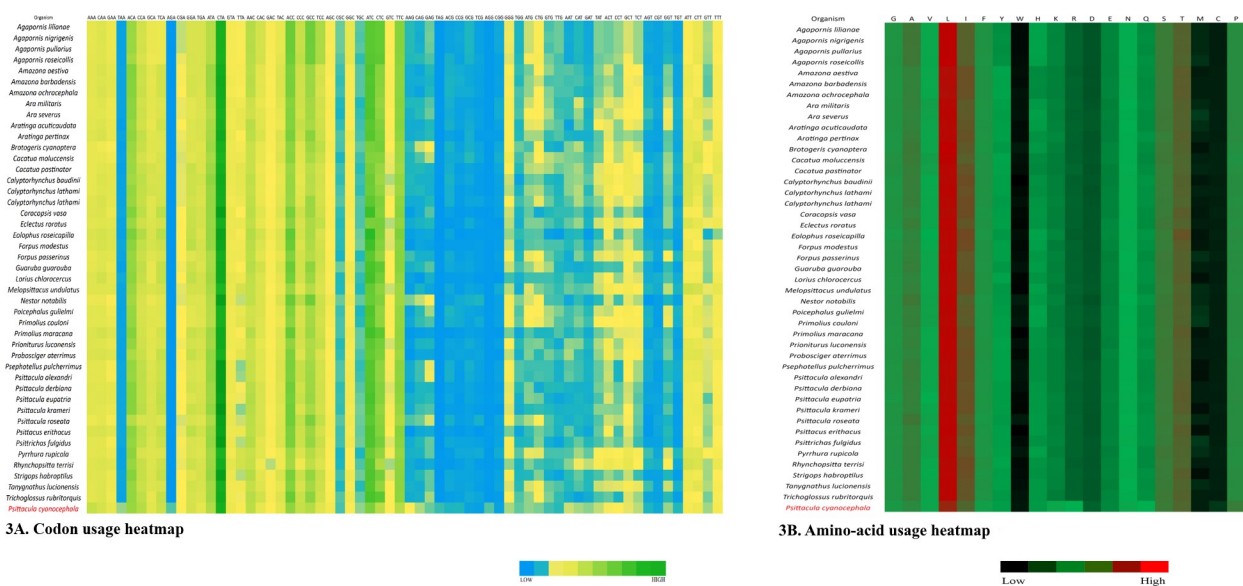

**3A. Codon usage heatmap**

**3B. Amino-acid usage heatmap**

**Fig 3.** Heatmaps based on (A) codon usage and (B) amino acid usage of all the 45 species compared in this study.

positions on amino acid acceptor arm, DU stem and TψC stem. The study found $trn^{Glu}$, and $trn^{Ser}$ (GCU) contained two wobble positions present on the amino acid acceptor arm and

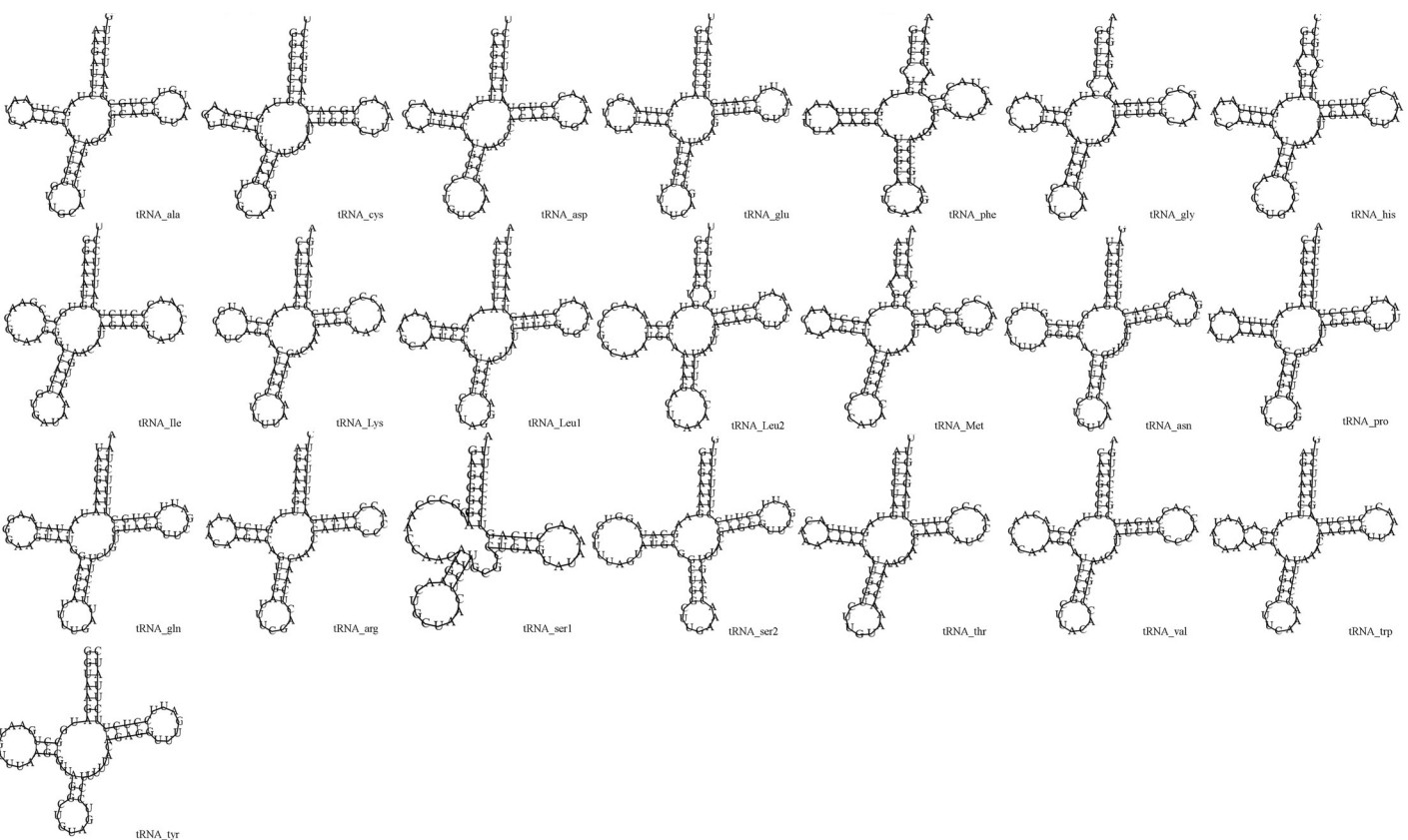

**Fig 4. Putative secondary structures of the 22 tRNA genes of *P. cyanocephala*.**

anticodon stem. Three tRNAs including $trn^{Ile}$, $trn^{Lys}$ and $trn^{Arg}$ contained no wobble positions. Rest of the tRNAs was found to have one wobble position each.

## Mitochondrial control region

Structurally, the D-loop of *P. cyanocephala* sequenced in this study consisted of 1317 base pairs similar to other birds, and located between $tRNA^{Glu}$ and $tRNA^{Phe}$. The D-loop consists of A (26.9%), C (26.4%), G (15.2%) and T (31.5%) indicating very high A+T content (Table 3). High A+T content was also noticed in the D-loop region of the other *Psittacula* species and in other birds [30, 51, 52].

The mitochondrial D-loop is a non-coding region mostly involved in initiation, replication, and regulation of transcription related activities in vertebrate mitogenomes [1]. The D-loop though hypervariable in sequence, is conserved in very selective fashion within or across various orders of vertebrates. As vertebrate mitochondrial genome is believed to evolve under strong selection pressure, various gene orders around the mitochondrial control region (CR) are believed to be dynamically inherited, duplicated, or degenerated during genome evolution [1]. *Psittaciformes* is a species-rich and taxonomically confusing taxa [1] and hence, several studies targeting the mitochondrial CR region of *Psittaciformes* have been undertaken to answer wider phylogenetic and taxonomic queries [50]. As such a typical (or ancestral) avian gene order varies from a typical vertebrate gene order [21]. However, previous studies have identified at least 7 re-arrangements of duplicated ancestral gene orders in some of the *Psittaciformes* species [1].

We report that the D-loop region of *P. cyanocephala* mitogenome displays the ancestral avian CR gene order. This typical ancestral gene order is also displayed by other *Psittacula* species (*P. alexandri*, *P. derbiana*, *P. eupatria*, *P. krameri* and *P. roseata)* compared in this study. It is believed that the duplicated CR region provides a selective advantage to size and energy metabolism efficiency [1]. Hence, parrots with duplicated CR regions can live long while supporting large body mass such as *Amazona* parrots [1]. Parakeets of *Psittacula* genus are small to medium-sized (~40 cm) and have a lower expected life span than that of the *Amazona* genus parrots. Such a life history trait can be explained by the presence of the ancestral CR gene order in *Psittacula* genera, assuming duplications and pseudogenization have resulted avian CR rearrangements [1, 51].

## Genetic divergence and phylogenetic analysis

The evolutionary rate of the vertebrate mitochondrial genome is rapid when compared to the nuclear genome while having a relatively stable genome structure and recombination ratio [50]. Hence, mitochondrial genome sequences are widely used to infer phylogenetic relationships as they offer small stable changes over a long period in any given taxa. In this regard, whole mitochondrial genomes relay better phylogenetic information than single-gene (nuclear/mitochondrial) phylogeny. To elucidate phylogenetic relationships in parrots/parakeets with a focus on *Psittacula* genus, nucleotide sequences of 13 PCGs concatenated from the whole mitochondrial genomes of 28 genera belonging to 6 parrot families were used in this study (Table 1).

The pairwise genetic divergence calculated for each individual genome and when grouped as a genus correlated well while clustering in the phylogenetic tree. The genus-level divergence analysis showed that the *Psittacula* genus was placed closest to *Tanygnathus* (0.08 units) then to *Eclectus* (0.09 units) and showed the highest divergence with *Agaporins*, *Rhynchopsitta*, *Primolius*, *Cacatua* and *Nestor* genus (0.19 units) (S4 Table). This result corroborates the previous report that *Psittacula* genus clusters with *Tanygnathus* and is closest to the *Eclectus* genus as a

parallel-group [8, 35]. The species-level (mitogenome based) divergence analysis elucidated that *P. cyanocephala* is closest to *P. roseata* (divergence of 0.06 units) than to any other *Psittacula* species (S5 Table). Divergence of *P. cyanocephala* with *P. eupatria*, *P. krameri*, *P. alexandri* and *P. derbiana* was calculated at 0.10, 0.12, 0.12 and 0.11 units respectively (S5 Table).

The phylogenetic tree obtained through both ML (Fig 5) and BI (S2 Fig) analysis reveals the same topologies for the *Psittacula* genus. Species were clustered in respective clades following their genus level separation. The genus *Eclectus*, *Psittacula*, *Tanygnathus* and *Prioniturus* have originated from a nearest common parrot ancestor. *P. cyanocephala* and *P. roseata* have branched together from a common node that runs parallel to other *Psittacula* and *Tanygnathus* species. As mentioned earlier this makes a strong case for the creation of monophyletic taxa of *P. cyanocephala* and *P. roseata*, which has been indicated by previous studies on single mitochondrial (*cob*)/nuclear (*rag-1*) markers [8, 14]. Another important fact evident from the clustering and divergence analysis is that *Tanygnathus* genus clusters within the *Psittacula* species, indicating that *Tanygnathus* and *Psittacula* species originated from a common ancestor.

A possible explanation for the phylogenetic incongruence of *Tanygnathus* and *Psittacula* generas reported in this study might have resulted from hybridization events i.e. introgression (admixture between genetic material of two species) and/or retention of ancestral polymorphisms because of incomplete lineage sorting during speciation (ILS) [57]. The phenomenon of introgression and ILS have been immensely appreciated in avian genomic studies and highly regarded as a pervasive mechanism of adaptive evolution, especially in birds [57]. Incongruent phylogenetic trees arising as a result of introgression/hybridization incidences are a common phenomenon. Avian orders are especially intriguing with 16% of the total bird species being affected by introgression [58]. Previous reports have attributed conflicting

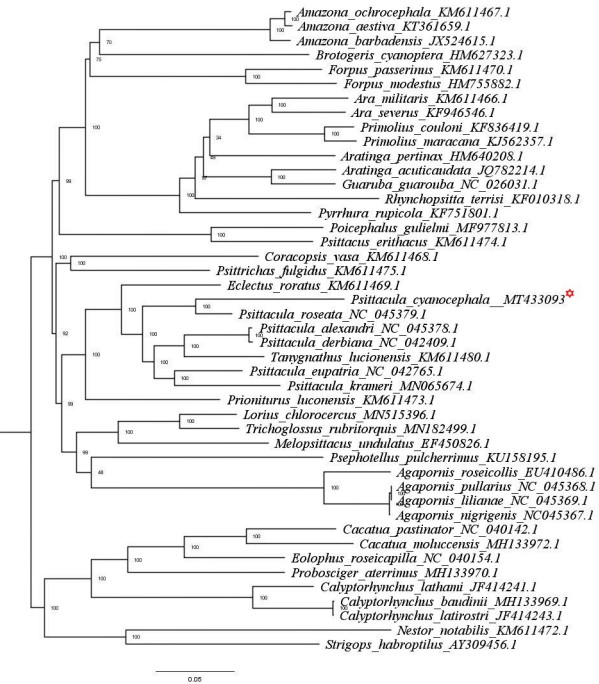

**Fig 5. ML tree based on the phylogenetic relationships of 45 *Psittaciformes* species determined using concatenated nucleotide sequences of 13 mitochondrial PCGs.** The tree was constructed in IQ-TREE employing TVM+F+G4 nucleotide substitution model, bootstrapped for 10000 replicates. *P. Cyanocephala* mitogenome is highlighted with red asterisk mark.

topologies in gene tree to introgression processes in woodpeckers [59], darwin's finches [60], flycatchers [20] and various other bird species [58]. On the other hand from a population point of view, ILS have been reported to affect the entire phylogeny of the Neoaves as well as the Palaeognathae clade, which have confounded the estimation of their respective species tree [61]. To summarize, speciation level events are notoriously hard to snapshot and hard bifurcations in gene trees difficult to distinguish against the phenomenon of introgression and/or ILS. Hence, the clustering of *Tanygnathus* genus within *Psittacula* complex may be attributed to the above mentioned population level incidences and the categorization of which is beyond the scope of our study [20, 61]. Different genomic entities (coding regions, non-coding regions, mitochondrial DNA, nuclear DNA) vary structurally, hence undergo different evolutionary constraints [61]. Phylogenetic analysis associated with varying genomic loci may produce discordance in their respective phylogenetic trees [20, 61]. If the genomic loci is under evolutionary constraint, the phylogenetic signal often gets masked with homoplasmy, multiple substitutions on the same site, and heterogeneous base composition [61]. In such cases whole mitogenomes presents itself as an excellent candidate for phylogenetic analysis. Ideally in absence of such evolutionary constraints (e.g. positive selection) mitochondrial DNA and W-linked loci (both female transmitted) are least less likely to introgress as compared to nuclear DNA loci (e.g. Z-linked loci) during hybridization [20, 61]. Previous studies have shown levels of mitochondrial DNA introgression to be nearly zero in avian hybrid zones, reiterating the fact that mitochondrial DNA are highly conserved across speciation events [20, 61]. In the most likelihood gene trees based on whole mitochondrial genome of *P. cyanocephala* is expected to ditto the true species phylogeny of *Psittacula* genus even in incidences of introgression and ILS.

From the pairwise divergence calculations, we observed that species belonging *Psittacula* genus are more closely related to *Tanygnathus* than to other *Psittacula* species. For instance *P. cyanocephala* is more closely related to *T. lucionensis* (0.11 units) than to *P. krameri* (0.12 units). Also, the phylogenetic trees obtained through ML and BI analyses display clustering of 4 separate groups within the trees. These groups include 1) *P. cyanocephala*, *P. roseata*; 2) *P. alexandri*, *P. derbiana*; 3) *T. lucionensis*, and 4) *P. eupatria*, *P. krameri*. These clustering pattern are supported by perfect nodal values (100%/100%) obtained in both ML and BI trees. Ambiguity in pairwise divergence scores and nestling of *Tanygnathus* within *Psittacula* genus complex makes a strong case for its taxonomic reconsideration. Previous studies of Braun [8] and Kundu [62] used morphological markers, genetic data and molecular dating approaches to present a near complete phylogeny of *Psittacula* genus. The unavailability of complete mitogenomes of all *Psittacula* and allied species, curtailed our data set and limited the scope of our phylogenetic analysis. However the phylogenetic results obtained in this study exactly mirror the incongruences presented by previous studies [8, 62]. Such genetic and evolutionary evidences weigh in support of *Psittacula* being is a non-monophyletic genus and warrants taxonomic reconsideration.

## Evolutionary analysis

dN/dS ratios were calculated to evaluate the effect of purifying selection via mutational pressure over protein-coding genes of *P. cyanocephala*. dN/dS values of all protein-coding genes were found to be less than 1 indicating the presence of purifying selection pressure on those genes (S6 Table). These values varied from 0.08 (*nad1*, *nad4l*) to 0.14 (*nad6*, *nad5* and *nad4*) suggesting differential selection constrains among genes. The lowest dN/dS values of *nad1* and *nad4l* indicated the strongest purifying selection, whereas, *nad6*, *nad5* and *nad4* were under

the least selection pressure. Thus, natural selection (purifying selection pressure) is acting as one of the major indices governing the evolution of *Psittaciformes*.

## Conclusion

We provide for the first time, the high-quality complete mitogenome of *P. cyanocephala*, a parakeet endemic to the Indian Subcontinent, whose existence is threatened by illegal bird trade. *P. cyanocephala* mitogenome consists of 37 genes and the gene arrangement shows conserved patterns like other parrots. We also report that D-loop region of *P. cyanocephala* mitogenome displays the ancestral avian CR gene order. Codon usage analysis revealed that third positions of the codons are dominated by either adenine or cytosine. The dN/dS results indicated that *nad1* and *nad4l* are under the strongest purifying selection whereas *nad6*, *nad5* and *nad4* are under the lower selection pressure in *P. cyanocephala*. Furthermore, phylogenetic results provided concrete evidence of multiple clades/groups clustering within the *Psittacula* genus supporting the call for a taxonomic reconsideration of the *Psittacula* genus. The complete mitogenome sequence of *P. cyanocephala* provided here will help in future phylogenetic and phylogeography studies as well as increase our understanding of the evolutionary relationships of such endemic species. Furthermore, mitogenome data will be instrumental in designing forensic tools for improving the conservation efforts and preventing the illegal trading of parakeet species.

## Supporting information

**S1 Table. PCR conditions for amplification of primers used for amplification of PCGs of mitogenome of *P. cyanocephala*.**
(DOCX)

**S2 Table. Sequence of primers and their predicted amplicon length used for amplification of various PCGs of mitogenome of *P. cyanocephala*.**
(DOCX)

**S3 Table. The Relative Synonymous Codon Usage (RSCU) values for each codon analysed in the 45 species of Psittaciformes order.** The optimal codons and their values are highlighted in the red font. Asterisk (*) indicates STOP codon. The amino acids are represented by their triple letter codes.
(XLSX)

**S4 Table. Pairwise distance values for 28 genera of Psittaciformes, where each genus is considered as a group for divergence analysis.** The divergence of Psittacula genus with respect to other genera is highlighted in red font.
(XLSX)

**S5 Table. Pairwise distance values for 45 species of Psittaciformes order, considered for divergence analysis in this study.** The divergence analysis values of Psittacula cyanocephala with respect to other species is highlighted in red font and a yellow background.
(XLSX)

**S6 Table. dN/dS values of *Psittaculacyanocephala* with respect to other parrot species for each Protein Coding Gene (PCG) is provided in the following table.** The Asterisk(*) / Numericals (1,2 .. 44) in the first row denotes asterisk (*) as *Psittacula cyanocephala* and the numbers as 44 species of *Psittaciformes* order analysed in this study. The numericals denote the following species: 1 *(Agapornis lilianae)*, 2 *(Agapornis nigrigenis)*,3 *(Agapornis pullarius)*, 4 *(Agapornis roseicollis)*, 5 *(Amazona aestiva)*, 6 *(Amazona barbadensis)*, 7 *(Amazona ochrocephala)*, 8 *(Ara militaris)*, 9 *(Ara severus)*, 10 *(Aratinga pertinax)*, 11 *(Brotogeris cyanoptera)*, 12

*(Cacatua moluccensis)*, 13 *(Cacatua pastinator)*, 14 *(Calyptorhynchus baudinii)*, 15 *(Calyptorhynchus lathami)*, 16 *(Calyptorhynchus latirostris)*, 17 *(Coracopsis vasa)*, 18 *(Eclectus roratus)*, 19 *(Eolophus roseicapilla)*, 20 *(Forpus modestus)*, 21 *(Forpus passerines)*, 22 *(Guaruba guarouba)*, 23 *(Lorius chlorocercus)*, 24 *(Melopsittacus undulates)*, 25 *(Nestor notabilis)*, 26 *(Poicephalus gulielmi)*, 27 *(Primolius couloni)*, 28 *(Primolius maracana)*, 29 *(Prioniturus lucionensis)*, 30 *(Probosciger aterrimus)*, 31 *(Psephotellus pulcherrimus)*, 32 *(Psittacula alexandri)*, 33 *(Psittacula derbiana)*, 34 *(Psittacula eupatria)*, 35 *(Psittacula krameri)*, 36 *(Psittacula roseata)*, 37 *(Psittacus erithacus)*, 38 *(Psittrichas fulgidus)*, 39 *(Pyrrhura rupicola)*, 40 *(Rhynchopsitta terrisi)*, 41 *(Strigops habroptilus)*, 42 *(Tanygnathus lucionensis)*, 43 *(Aratinga acuticaudata)*, 44 *(Trichoglossus rubritorquis)*.
(XLSX)

**S1 Fig. Gene arrangement of all the 45 species of *Psittaciformes* family compared in this study.** The abbreviations are as follows: ali *(Agapornis lilianae)*, ani *(Agapornis nigrigenis)*,apu *(Agapornis pullarius)*, aro *(Agapornis roseicollis)*, aaest *(Amazona aestiva)*, abarb *(Amazona barbadensis)*, aochr *(Amazona ochrocephala)*, amilitari *(Ara militaris)*, as *(Ara severus)*, aacu *(Aratinga acuticaudata)*, aper *(Aratinga pertinax)*, boc *(Brotogeris cyanoptera)*, cmolu *(Cacatua moluccensis)*, cpast *(Cacatua pastinator)*, cbau *(Calyptorhynchus baudinii)*, clath *(Calyptorhynchus lathami)*, clatir *(Calyptorhynchus latirostris)*, cvasa *(Coracopsis vasa)*, eror *(Eclectus roratus)*, eros *(Eolophus roseicapilla)*, fomo *(Forpus modestus)*, fopa *(Forpus passerines)*, gugu *(Guaruba guarouba)*, lochlo *(Lorius chlorocercus)*, mun *(Melopsittacus undulates)*, neno *(Nestor notabilis)*, pogu *(Poicephalus gulielmi)*, prico *(Primolius couloni)*, prima *(Primolius maracana)*, pluco *(Prioniturus lucionensis)*,pater *(Probosciger aterrimus)*, psepul *(Psephotellus pulcherrimus)*, palex *(Psittacula alexandri)*, pderb *(Psittacula derbiana)*, peup *(Psittacula eupatria)*, pkra *(Psittacula krameri)*, pro *(Psittacula roseata)*, peri *(Psittacus erithacus)*, pful *(Psittrichas fulgidus)*, prup *(Pyrrhura rupicola)*, rhyter *(Rhynchopsitta terrisi)*, strihab *(Strigops habroptilus)*, tluci *(Tanygnathus lucionensis)*, trub *(Trichoglossus rubritorquis)*, pcyano *(Psittacula cyanocephala)*.
(DOCX)

**S2 Fig. BI tree based on the phylogenetic relationships of 45 *Psittaciformes* species determined using concatenated nucleotide sequences of 13 mitochondrial PCGs.** The tree was constructed in Mr.Bayes employing GTR+I+G nucleotide substitution model following 4 independent chains running for 100,000 generations, sub-sampling every 1000 generations and using a burn-in of 100 generations. *P. Cyanocephala* mitogenome is highlighted with red asterisk mark.
(DOCX)

**S1 File.**
(RAR)

## Acknowledgments

We are also thankful to the Principal Chief Conservator of Forest (Wildlife), Government of Maharashtra for providing the permissions to obtain biological samples from Transit Treatment Centre, Nagpur, Maharashtra. We are thankful to Dr. Syed Bilal Ali, veterinarian at Transit Treatment Centre, Nagpur, Maharashtra for collecting biological samples.

## Author Contributions

**Conceptualization:** Venkata Hanumat Sastry Kochiganti, Goldin Quadros, Ram Pratap Singh.

**Data curation:** Indrani Sarkar, Saurabh Singh Rathore, Vikram Singh.

**Formal analysis:** Prateek Dey, Indrani Sarkar, Saurabh Singh Rathore, Vikram Singh.

**Funding acquisition:** Padmanabhan Pramod, Ram Pratap Singh.

**Investigation:** Prateek Dey, Sanjeev Kumar Sharma, Indrani Sarkar, Ram Pratap Singh.

**Methodology:** Prateek Dey, Sanjeev Kumar Sharma, Swapna Devi Ray, Ram Pratap Singh.

**Project administration:** Ram Pratap Singh.

**Software:** Prateek Dey, Indrani Sarkar.

**Supervision:** Ram Pratap Singh.

**Validation:** Prateek Dey.

**Visualization:** Indrani Sarkar.

**Writing – original draft:** Prateek Dey, Ram Pratap Singh.

**Writing – review & editing:** Prateek Dey, Sanjeev Kumar Sharma, Goldin Quadros, Saurabh Singh Rathore, Vikram Singh, Ram Pratap Singh.

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
