## [Decision Letter · Decision Letter 0]

14 Dec 2020

PONE-D-20-31563

Complete mitogenome of endemic Plum-headed parakeet Psittacula cyanocephala – characterization and phylogenetic analysis

PLOS ONE

Dear Dr. SINGH,

Thank you for submitting your manuscript to PLOS ONE. After careful consideration, we feel that it has merit but does not fully meet PLOS ONE’s publication criteria as it currently stands. Therefore, we invite you to submit a revised version of the manuscript that addresses the points raised during the review process.

We look forward to receiving your revised manuscript.

Kind regards,

Maria Andreína Pacheco, Ph.D.

Academic Editor

PLOS ONE

Additional Editor Comments (if provided):

Please, address the issues that both reviewers suggested.

Journal Requirements:

2) In your Methods section, please state the volume of the blood samples collected for use in your study.

3) We note that you are reporting an analysis of a microarray, next-generation sequencing, or deep sequencing data set. PLOS requires that authors comply with field-specific standards for preparation, recording, and deposition of data in repositories appropriate to their field. Please upload these data to a stable, public repository (such as ArrayExpress, Gene Expression Omnibus (GEO), DNA Data Bank of Japan (DDBJ), NCBI GenBank, NCBI Sequence Read Archive, or EMBL Nucleotide Sequence Database (ENA)). In your revised cover letter, please provide the relevant accession numbers that may be used to access these data. For a full list of recommended repositories, see http://journals.plos.org/plosone/s/data-availability#loc-omics or http://journals.plos.org/plosone/s/data-availability#loc-sequencing.

4) PLOS requires an ORCID iD for the corresponding author in Editorial Manager on papers submitted after December 6th, 2016. Please ensure that you have an ORCID iD and that it is validated in Editorial Manager. To do this, go to ‘Update my Information’ (in the upper left-hand corner of the main menu), and click on the Fetch/Validate link next to the ORCID field. This will take you to the ORCID site and allow you to create a new iD or authenticate a pre-existing iD in Editorial Manager. Please see the following video for instructions on linking an ORCID iD to your Editorial Manager account: https://www.youtube.com/watch?v=_xcclfuvtxQ

Reviewers' comments:

Reviewer's Responses to Questions

**Comments to the Author**

1. Is the manuscript technically sound, and do the data support the conclusions?

Reviewer #1: Partly

Reviewer #2: Partly

2. Has the statistical analysis been performed appropriately and rigorously? 

Reviewer #1: Yes

Reviewer #2: Yes

3. Have the authors made all data underlying the findings in their manuscript fully available?

Reviewer #1: Yes

Reviewer #2: Yes

4. Is the manuscript presented in an intelligible fashion and written in standard English?

Reviewer #1: No

Reviewer #2: No

5. Review Comments to the Author

Reviewer #1: The most important results obtained by the authors of the manuscript: “Complete mitogenome of endemic Plum-headed parakeet Psittacula cyanocephala – characterization and phylogenetic analysis” concern the following aspects:

1. Amplification and sequencing of the first Psittacula cyanocephala mitogenome.

2. Its comparison with other mitogenomes representative for avian order Psittaciformes. For this purpose, the authors performed a phylogenetic analysis, determined the pairwise genetic divergence and calculated dN/dS ratios for PCGs.

The impact of the obtained results on the understanding of the taxonomy of the genus Psittacula and Tanygnathus was summarized by the authors in the section conclusion:

“Furthermore, phylogenetic results provided concrete evidence of multiple phyla clustering within the Psittacula genus supporting the call for a taxonomic reconsideration of the Psittacula genus.”

My comments:

I fully agree with the authors of the manuscript that Psittacula cyanocephala mitogenome will be very useful in future phylogenetic and phylogeographic studies, which will allow to understand the evolution of Tanygnathus and Psittacula genera.

But actually …. Phylogenetic analysis presented in this manuscript did not bring anything new. The placement of Psittacula cyanocephala in the clade of roseata/cyanocephala/himlayana/finschi species is commonly known (Kundu et al., 2012; Podsiadłowski et al., 2017; Braun et al., 2019). Moreover, the same publications showed relations between Psittacula groups and Tanygnathus species, which are identical with those presented in the manuscript. Additionally, Braun et al., proposed many taxonomical changes based on obtained results.

In general, anything new from the phylogenetic and taxonomic view is presented in this manuscript. Of course, I fully understand the intention of the authors who wanted to check whether the results obtained from the complete mitochondrial genomes will be consistent with the Braun`s results obtained with the use of only two quite short sequences: mitochondrial cytb and nuclear RAG-1 genes. However, it is impossible to obtain an irrefutable conclusion with the use of one Tanygnathus and six Psittacula mitogenomes, which even do not represent all “taxonomical” clades being considered by Braun. The fact, that Psittacula genus should be taxonomical revised is indisputable but commonly known.

The answer for the question “if taxonomical revision proposed by Braun et al., 2019 is proper” should be the real scientific novelty of this manuscript.

Unfortunately, results presented in this manuscript are only the information about new Psittacula mitogenome and its analysis. These results should be published in journals dedicated to such preliminary analyses. In my opinion, their scientific novelty is not sufficient to be published in PLOS ONE Journal.

Reviewer #2: In this manuscript, the authors aim to describe the complete sequence of the mitochondrial genome of Psittacula cyanocephala. They report general and comparative statistics about the mitochondrial DNA of this species as compared to other related species. Lastly, they estimate a phylogenetic tree based on the protein coding genes and suggest a taxonomic revision of the Psittacula genus based on it. The authors need to address some issues, which are described below, before the paper is suitable for publication.

Major issue

- Phylogenetic results.

The claim that the genus needs a taxonomic revision based on their findings is weak. First, the placement of Tanygnathus together with the Psittacula species may be due to other causes that were not discussed (such as introgression and ILS) by the authors. This should be explicitly addressed (and discussed) in the text. Second, it is not clear in the text what would be the four clusters inside the genus and why they are relevant (the authors use this argument to justify the taxonomic reconsideration). It is mentioned in the abstract that “Phylogenetic analyses revealed the Psittacula genus as paraphyletic nature, containing at least 4 groups of species within the same genus, suggesting its taxonomic reconsideration”. I do not see why this is an argument for taxonomic revision. Please expand the discussion about the phylogenetic results addressing the possible causes for the recovered phylogenetic relationships and provide stronger arguments for a taxonomic revision.

Minor issues:

- English: I recommend that authors seek a scientific editing service to improve the quality of the English.

- Figures: I suggest increasing the font size of the figures.

- Lines 72-74: Braun and coworkers did not use “differences in nucleotide sequences” to reconstruct a phylogenetic tree for Psittacula. They used a ML approach based on a substitution model, which was HKY. Please correct.

- Lines 77-78: Please provide arguments that justify your claim. Why does complete mitochondrial genome data is the best option instead of large amounts of nuclear data?

- Lines 87-89: The work cited is about mammals. Please include a citation for birds, or explicitly state “…provide consistent results compared to nuclear genes for mammals” and justify why the same would be expected for birds.

- Line 148: The select protein coding sequences that were used to verify the mitogenome should me mentioned at this point.

- Lines 194-198: Please provide more details about the ML approach used to estimate dN/dS (parameters, method…).

- Line 203: “The number of base substitutions per site can provide…”. I suggest the use of the “pairwise base substitutions” terminology.

- Lines 205-206: “Both the genome-based and genus-based divergence analysis was performed”. I suggest changing “genome-based” and “genus-based” to “species-level” and “genus-level”, respectively. Additionally, it is advisable to explain why these two distinct approaches were considered.

- Line 207: “members of the same genus were considered as one group”. Explain how this was done (average?).

- Line 208: I suggest rephrasing the sentence to something similar to “Thus all considered organisms were clustered into 28 different groups, which corresponds to the 28 genera”.

- Line 212: Please explain what “emphasis” mean. State the criteria to choose the sequences/genus to keep along the phylogenetic analyses.

- Line 214: Please state if the sequences were aligned independently by gene and based on the amino acid sequence, which is desirable.

- Lines 219-220: Please state what “effective GTR+I+G” means (I am assuming this is somehow different from the regular GTR+I+G, otherwise the word “effective” would not be necessary).

- Lines 220-221: Please provide what statistics were used to check the convergence and mixing of the Bayesian chains.

- Lines 228-229: How did the authors deal with the similarity between the sequences generated by Sanger sequencing and NGS? It seems like 99% of similarity was considered enough, but it should be explicitly mentioned.

- Line 245: Please change “lower” to “upper”.

- Lines 283-285: This sentence is confusing. What does “in nature” mean?

- Lines 304-305: Please explain and discuss why Fig 2E provide evidence for selection pressure.

- Lines 321-322: Please say the meaning of the black squares in Fig 3B.

- Lines 380-381: You can only say that if you assume that the hypothesis of duplicated CRs is correct.

- Line 387: Here, by “single-gene” you mean mitochondrial gene or any gene (nuclear and mitochondrial)?

- Line 389: This sentence is confusing. How can whole mitochondrial genomes be used along with concatenated PCGs? The PCGs are part of the mitogenome.

- Line 398: I suggest rephrasing this sentence to “The mitogenome-based divergence analysis suggest that…”

- Lines 402-404: I did not understand what topology was presented before.

- Lines 408-409: It would be reasonable to mention what kind of data the cited works were based on.

- Lines 409-411: This indicates that the mitochondrial DNA of this species have common ancestry. This pattern does not necessarily imply a paraphyletic genus. It may be caused by populational level phenomena such as introgression.

- Lines 415-419: All these statements are assuming that the mitogenome is always identical to the species phylogeny, which is not true. Also, this part of the text is very confusing. I was not able to understand why there are non-monophiletic clustering os 4 groups.

- Lines 424-433: This analysis is pointless, unless the results are discussed and compared to other birds/vertebrates. Besides that, are the dN/dS values statistically significant?

- Lines 441-443: Change “least” to “lower”.

- Line 444: Change “multiple phyla” to “multiple clades/groups”.

- Line 446: Change “phylogeography” to “phylogeographic”.

6. PLOS authors have the option to publish the peer review history of their article (what does this mean?). If published, this will include your full peer review and any attached files.

Reviewer #1: No

Reviewer #2: No

---

## [Author Response · Author response to Decision Letter 0]

23 Jan 2021

Ms. Ref. No.: PONE-D-20-31563 - EMID:75da62b30f6ba005

Title: Complete mitogenome of endemic Plum-headed parakeet Psittacula cyanocephala – characterization and phylogenetic analysis

Response to reviewers

We would like to thank Dr. Maria Andreína Pacheco (Academic Editor) and the Reviewers for their constructive remarks, which were of great help to improve this manuscript. We have revised the manuscript as suggested by the Reviewers. The revised manuscript is submitted for further consideration. Please find below a point-by-point rebuttal to the issues raised.

Additional Editor Comments:

Comment 1: Please address the issues that both reviewers suggested. 

Response: We have addresses all the issues raised by both reviewers. 

Reviewer #1

Comment 1: I fully agree with the authors of the manuscript that Psittacula cyanocephala mitogenome will be very useful in future phylogenetic and phylogeographic studies, which will allow to understand the evolution of Tanygnathus and Psittacula genera. 

But actually …. Phylogenetic analysis presented in this manuscript did not bring anything new. The placement of Psittacula cyanocephala in the clade of roseata/cyanocephala/himlayana/finschi species is commonly known (Kundu et al., 2012; Podsiad?owski et al., 2017; Braun et al., 2019). Moreover, the same publications showed relations between Psittacula groups and Tanygnathus species, which are identical with those presented in the manuscript. Additionally, Braun et al., proposed many taxonomical changes based on obtained results. 

In general, anything new from the phylogenetic and taxonomic view is presented in this manuscript. Of course, I fully understand the intention of the authors who wanted to check whether the results obtained from the complete mitochondrial genomes will be consistent with the Braun`s results obtained with the use of only two quite short sequences: mitochondrial cytb and nuclear RAG-1 genes. However, it is impossible to obtain an irrefutable conclusion with the use of one Tanygnathus and six Psittacula mitogenomes, which even do not represent all “taxonomical” clades being considered by Braun. The fact, that Psittacula genus should be taxonomical revised is indisputable but commonly known. 

The answer for the question “if taxonomical revision proposed by Braun et al., 2019 is proper” should be the real scientific novelty of this manuscript. 

Unfortunately, results presented in this manuscript are only the information about new Psittacula mitogenome and its analysis. These results should be published in journals dedicated to such preliminary analyses. In my opinion, their scientific novelty is not sufficient to be published in PLOS ONE Journal.

Response: Thank you very much for your thoughtful comments and insight. We performed phylogenetic analysis using whole mitochondrial genomes of Psittacula genus to elucidate phylogenetic inconsistencies within the genus and supported call for its taxonomic reconsideration as suggested by previous studies (Braun et al., 2019; Kundu et al., 2012). We understand that incongruent clustering of different clades within Psittacula genus has been reported by previous studies using single mitochondrial/nuclear genes (Braun et al., 2019; Kundu et al., 2012), hence we aimed at using complete mitogenomes to verify these claims as you pointed out correctly. And through this study we have successfully verified such claims and presented a more comprehensive phylogenetic analysis using complete mitogenomes, than the previous studies.

However, we understand that in absence of complete mitogenomes of all “taxonomical” members of Psittacula genus, recovering an exact and irrefutable phylogenetic tree is impossible. However, we have made efforts to explain in our text {Introduction section (please see lines 78-94 on page 4 of file named ‘Manuscript’); Results and Discussion section (please see lines 440-469 on page 23 of file named ‘Manuscript’)} as why the complete mitochondrial genomes are an indispensable tool to achieve a plausible phylogenetic tree in such cases of taxonomic confusion. Absence of data or missing members of certain “taxonomic clades”, may hinder but definitely not forfeit the quest for a better phylogenetic understanding of Psittacula genus as compared to previously published studies. And through our study we have strived to achieve this scientific understanding. Hence, we believe our results important and re-verified known facts through superior comprehensive approaches, providing baseline data for further analysis on phylogenetic discordances within the Psittacula genus. 

Reviewer #2

Major issue: The claim that the genus needs a taxonomic revision based on their findings is weak. First, the placement of Tanygnathus together with the Psittacula species may be due to other causes that were not discussed (such as introgression and ILS) by the authors. This should be explicitly addressed (and discussed) in the text. Second, it is not clear in the text what would be the four clusters inside the genus and why they are relevant (the authors use this argument to justify the taxonomic reconsideration). It is mentioned in the abstract that “Phylogenetic analyses revealed the Psittacula genus as paraphyletic nature, containing at least 4 groups of species within the same genus, suggesting its taxonomic reconsideration”. I do not see why this is an argument for taxonomic revision. Please expand the discussion about the phylogenetic results addressing the possible causes for the recovered phylogenetic relationships and provide stronger arguments for a taxonomic revision.

Response: Our gratitude for your critical comments and valuable suggestion.

We agree the findings on taxonomic incongruence of Psittacula and Tanygnathus genus may have arisen due to introgression and incomplete lineage sorting (ILS), which has been addressed and discussed in the Results and Discussion section in a detailed way. Incongruent phylogenetic trees arising as a result of introgression/hybridization incidences are a common phenomenon with 16% of the total bird species being affected (Otthenburghs et al. 2015; Otthenburghs et al. 2017) such as woodpeckers (Fuchs et al. 2013), darwin’s finches (Lamichhaney et al. 2018), flycatchers (Rheindt and Edwards, 2011) and various other bird species (Otthenburghs et al. 2015). On the other hand from a population point of view, ILS have been reported to affect the entire phylogeny of the Neoaves as well as the Palaeognathae clade, which have confounded the estimation of their respective species tree (Stiller and Zhang, 2019). Hence, the clustering of Tanygnathus genus within Psittacula complex may be attributed to the above mentioned population level incidences and the categorization of which is beyond the scope of our study (Rheindt and Edwards, 2011; Stiller and Zhang, 2019). Please see lines 437-454 on page 23 of file named ‘Manuscript’.

For, the second part of the comment we have explicitly addressed the 4 “taxonomic clusters” inside the genus that include 1) P. cyanocephala, P. roseata; 2) P. alexandri, P. derbiana; 3) T. lucionensis, and 4) P. eupatria, P. krameri. Evidences from the pairwise divergence calculations, pointed that species belonging Psittacula genus (e.g. P. cyanocephala) are more closely related to Tanygnathus than to other members Psittacula genus. Thus clustering pattern of these groups are supported by perfect nodal values (100%/100%) obtained in both ML and BI trees. Previously, Braun (Braun et al. 2019) and Kundu (Kundu et al. 2012) used morphological markers, genetic data and molecular dating approaches to present a near complete phylogeny of Psittacula genus. Evidences from pairwise divergence scores, strong nodal values in phylogenetic trees for discordant branches and mirroring of incongruences presented by previous studies weigh in support of Psittacula being a non-monophyletic genus and warrants taxonomic reconsideration. Please see lines 470-485 on page 23 of file named ‘Manuscript’.

• Ottenburghs J, Kraus RH, van Hooft P, van Wieren SE, Ydenberg RC, Prins HH. Avian introgression in the genomic era. Avian Res. 2017 Dec 1;8(1):30.

• Ottenburghs J, Ydenberg RC, Van Hooft P, Van Wieren SE, Prins HH. The Avian Hybrids Project: gathering the scientific literature on avian hybridization. Ibis. 2015 Oct;157(4):892-4.

• Lamichhaney S, Han F, Webster MT, Andersson L, Grant BR, Grant PR. Rapid hybrid speciation in Darwin’s finches. Science. 2018 Jan 12;359(6372):224-8.

• Fuchs J, Pons JM, Liu L, Ericson PG, Couloux A, Pasquet E. A multi-locus phylogeny suggests an ancient hybridization event between Campephilus and melanerpine woodpeckers (Aves: Picidae). Mol. Phylogenet. Evol. 2013 Jun 1;67(3):578-88.

• Rheindt FE, Edwards SV. Genetic introgression: an integral but neglected component of speciation in birds. The Auk. 2011 Oct 1;128(4):620-32.

• Stiller J, Zhang G. Comparative phylogenomics, a stepping stone for bird biodiversity studies. Diversity. 2019 Jul;11(7):115.

• Braun MP, Datzmann T, Arndt T, Reinschmidt M, Schnitker H, Bahr N. A molecular phylogeny of the genus Psittacula sensulato (Aves: Psittaciformes: Psittacidae: Psittacula, Psittinus, Tanygnathus,†Mascarinus) with taxonomic implications. Zootaxa. 2019 March;4563(3):547-562.

• Kundu S, Jones CG, Prys-Jones RP, Groombridge JJ. The evolution of the Indian Ocean parrots (Psittaciformes): extinction, adaptive radiation and eustacy. Mol. Phylogenet. Evol. 2012 Jan 1;62(1):296-305.

Minor issues:

Comment 1. English: I recommend that authors seek a scientific editing service to improve the quality of the English.

Response: We are unable to take help from scientific editing service due to paucity of funds. However, the quality of English of the manuscript has been improved. 

Comment 2. Figures: I suggest increasing the font size of the figures.

Response: The font size in the figures has been adjusted to the maximum to maintain quality and structural integrity of the images.

Comment 3. Lines 72-74: Braun and coworkers did not use “differences in nucleotide sequences” to reconstruct a phylogenetic tree for Psittacula. They used a ML approach based on a substitution model, which was HKY. Please correct.

Response: Phrases corrected in the text to mention Braun and coworkers reconstructed used ML based tree with HKY model. Confusion is regretted. Please see lines 71-74 on page 4 of file named ‘Manuscript’.

Comment 4. Lines 77-78: Please provide arguments that justify your claim. Why does complete mitochondrial genome data is the best option instead of large amounts of nuclear data?

Response: Complete mitochondrial genomes yields more species accurate phylogenetic tree as compared to large amounts of nuclear data (Campbell and Lapointe, 2011; Rheindt and Edwards, 2011). Mitochondrial genomes are comparatively more conserved than nuclear genomes during transition events and its unique architecture provides mitochondrial genomes the ability to carry phylogenetic information more consistently (Rheindt and Edwards, 2011). Even in incidences of introgression or ILS, mitochondrial DNA has proved to be more conserved then nuclear DNA thus avoiding stochastic errors of homoplasmy and heterogeneous base composition during phylogenetic analysis (Stiller and Zhang, 2019). Overall whole mitogenomes presents itself as an excellent candidate for phylogenetic analysis and such a fact has been recorded by previous workers in multiple studies (Rheindt and Edwards, 2011; Stiller and Zhang, 2019). Please see lines 78-96 and 454-471 on page 4 and 23 respectively of file named ‘Manuscript’.

• Campbell V, Lapointe FJ. Retrieving a mitogenomic mammal tree using composite taxa. Mol. Phylogenet. Evol. 2011 Feb 1;58(2):149-56.

• Rheindt FE, Edwards SV. Genetic introgression: an integral but neglected component of speciation in birds. The Auk. 2011 Oct 1;128(4):620-32.

• Stiller J, Zhang G. Comparative phylogenomics, a stepping stone for bird biodiversity studies. Diversity. 2019 Jul;11(7):115.

Comment 5. Lines 87-89: The work cited is about mammals. Please include a citation for birds, or explicitly state “provide consistent results compared to nuclear genes for mammals” and justify why the same would be expected for birds

Response: New citation for birds has been added as Reference No. 20. Please see lines 91-93 on page 4 of file named ‘Manuscript’.

• Rheindt FE, Edwards SV. Genetic introgression: an integral but neglected component of speciation in birds. The Auk. 2011 Oct 1;128(4):620-32.

Comment 6. Line 148: The select protein coding sequences that were used to verify the mitogenome should me mentioned at this point.

Response: The select protein coding genes of cox1, cox2, atp8, atp6, nad1, nad2, nd3 and cob have been mentioned in the text, which were used to verify the mitogenome. Please see lines 150-152 on page 7 of file named ‘Manuscript’.

Comment 7. Lines 194-198: Please provide more details about the ML approach used to estimate dN/dS (parameters, method…). 

Response: Additional details about parameters and methods used for the ML approach used to estimate dN/dS ratio has been provided in the Materials and Methods section of the text. PAL2NAL program embedded in Phylogenetic Analysis by Maximum Likelihood (PAML) package was used for this analysis. We have exploited the standalone Linux version of this server. The codons used in mRNAs were aligned and was subjected to dN/dS analysis. The orthologus pair of gene sequences was given as input file and the dN/dS value for that gene was calculated. Please see lines 200-206 on page 10 of file named ‘Manuscript’.

Comment 8. Line 203: “The number of base substitutions per site can provide…”. I suggest the use of the “pairwise base substitutions” terminology.

Response: Terminology changed. Pairwise base substitutions terminology is used hence further in this manuscript. Please see lines 210-212 on page 10 of file named ‘Manuscript’.

Comment 9. Lines 205-206: “Both the genome-based and genus-based divergence analysis was performed”. I suggest changing “genome-based” and “genus-based” to “species-level” and “genus-level”, respectively. Additionally, it is advisable to explain why these two distinct approaches were considered.

Response: The terminologies changed as suggested to the advised terms in the manuscript and further denoted as ‘genus-level’ and ‘species-level’. The reason for adopting two distinct approaches stems for the need to calculate difference between each individual species, where a genetic distance between members of even the same genus can be studied. Whereas studying the average distance between different generas by clubbing multiple mitogenomes as a group, helps us to observe as a genus, which all are closely related and which distantly. Overall, such partitioning patterns help us to understand the genetic divergence indices in a comprehensive way. Please see lines 214-219 on page 11 of file named ‘Manuscript’.

Comment 10. Line 207: “members of the same genus were considered as one group”. Explain how this was done (average?).

Response: Members of the genus were partitioned into groups in MEGA X, wherein the average distances between the genus-level groups were evaluated apart from species-level genetic distances of individual mitogenomes. We followed the pre-embedded programs of MEGA X for partitioning data into groups and evaluating distances within groups. Please see lines 214-217 on page 11 of file named ‘Manuscript’.

Comment 11. Line 208: I suggest rephrasing the sentence to something similar to “Thus all considered organisms were clustered into 28 different groups, which corresponds to the 28 genera”.

Response: Sentence rephrased in the manuscript. Please see lines 219-220 on page 11 of file named ‘Manuscript’.

Comment 12. Line 212: Please explain what “emphasis” mean. State the criteria to choose the sequences/genus to keep along the phylogenetic analyses.

Response: Emphasis means, only the results from the Psittacula, Tanygnathus and Eclectus generas presented in the phylogenetic tree were explained in detail in the Results section. ‘Emphasis’ doesn’t imply any specific statistical weightage given to the above mentioned groups during analysis. Please see lines 219-220 on page 11 of file named ‘Manuscript’.

All available members of Psittaciformes order, belonging to 28 generas and 45 different species were used for phylogenetic analysis. Only one mitogenome from each species (as specified in NCBI-Genbank) were taken for further analysis. By including 45 species, and higher number of out-groups we aimed to construct a comprehensive phylogeny from whole mitogenome of Psittaciformes for the first time. This would in turn show the true position of members of Psittacula genus within the Psittaciformes order in a holistic way. Please see Table. 1 on page 8-9 of file named ‘Manuscript’.

Comment 13. Line 214: Please state if the sequences were aligned independently by gene and based on the amino acid sequence, which is desirable.

Response: The nucleotide sequences were concatenated to the protein coding regions only. These sequences (containing only protein coding genes) were then aligned using MUSCLE. Please see lines 225-226 on page 11 of file named ‘Manuscript’.

Comment 14. Lines 219-220: Please state what “effective GTR+I+G” means (I am assuming this is somehow different from the regular GTR+I+G, otherwise the word “effective” would not be necessary).

Response: Apologies for the confusion. The phrase meant to mean the same GTR+I+G model. The word ‘effective’ in this context has been deleted from the manuscript to increase clarity. Please see lines 231-232 on page 11 of file named ‘Manuscript’.

Comment 15. Lines 220-221: Please provide what statistics were used to check the convergence and mixing of the Bayesian chains.

Response: No statistical tests were used to check the convergence and mixing of Bayesian chains. We trusted that proper model selection along with very high posterior values (~100/100) obtained in the analysis, to provide accurate results. Results from the Bayesian analysis dittoed the tree obtained through ML analysis, thus re-affirming our assumption. Please see the nodal values in Fig. 5 and Supplementary Figure ‘S2 Fig’.

Comment 16. Lines 228-229: How did the authors deal with the similarity between the sequences generated by Sanger sequencing and NGS? It seems like 99% of similarity was considered enough, but it should be explicitly mentioned.

Response: The sequences were expected to be completely similar, however owing to lower confidence of base-calls in certain Sanger sequences, we decide to adopt 99% as considerable score, enough for sequence verification. Quality of base-calls lowered owing to difficulty in sequencing due to possible secondary structures in the template. Please see lines 240-242 on page 12 of file named ‘Manuscript’ where the criteria is mentioned explicitly.

Comment 17. Line 245: Please change “lower” to “upper”.

Response: Apologies for the error. Changed “lower” to “upper” in the manuscript.

Comment 18. Lines 283-285: This sentence is confusing. What does “in nature” mean?

Response: The phrase “in nature” was used as a discourse marker. Regrets for the mistake, the phrase has been deleted and sentence made more legible. Please see lines 295-297 on page 17 of file named ‘Manuscript’.

Comment 19. Lines 304-305: Please explain and discuss why Fig 2E provide evidence for selection pressure.

Response: The effective number of codons (ENc) primarily reflects the mutational bias. Plotting ENc against the GC3 thus provide an evident of whether mutational pressure or natural selection is acting on the protein coding gens of an organism. An umbrella-line is drawn with the “expected ENc value” (value assuming only mutational pressure is acting on considered genes) and is compared to the observed ENc values. Several previous reports including Roy et al. 2015, Smith 2019 (Enhanced effective codon numbers to understand codon usage bias) have stated that, if the observed ENc value exceeds the expected ENc value it will depict the complete mutational pressure on those genes. However, if the observed value is less than the expected value it is due to the selection pressure lowering the effective number of codons. In this figure, all the plotted points have placed below the umbrella-line indicating the selection pressure over mutational bias on those genes. Please see lines 317-328 on page 18 of file named ‘Manuscript’.

• Roy A, Mukhopadhyay S, Sarkar I, Sen A. Comparative investigation of the various determinants that influence the codon and amino acid usage patterns in the genus Bifidobacterium. World. J. Microb. Biot. 2015 Jun 1;31(6):959-81. 

• Smith R. Enhanced effective codon numbers to understand codon usage bias. BioRxiv. 2019 Jan 1:644609.

Comment 20. Lines 321-322: Please say the meaning of the black squares in Fig 3B.

Response: Our apologies for the mistake, colour code has been added in the figure Fig. 3B. Black squares denote the lowest values on the amino acid heatmap.

Comment 21. Lines 380-381: You can only say that if you assume that the hypothesis of duplicated CRs is correct.

Response: The statement can be made if only the hypothesis of duplicated CRs is correct, hence we have rephrased the sentence. Multiple previous studies have observed duplicated CR region has contributed to longevity in birds (Sarkar et al. 2020; Skujina et al. 2019). Please see lines 402-404 on page 21 of file named ‘Manuscript’.

• Skujina I, McMahon R, Lenis VP, Gkoutos GV, Hegarty M. Duplication of the mitochondrial control region is associated with increased longevity in birds. Aging (Albany NY). 2016 Aug;8(8):1781.

• Sarkar I, Dey P, Sharma SK, Ray SD, Kochiganti VH, Singh R, Pramod P, Singh RP. Turdoides affinis mitogenome reveals the translational efficiency and importance of NADH dehydrogenase complex-I in the Leiothrichidae family. Sci. Rep. 2020 Oct 1;10(1):1-1.

Comment 22. Line 387: Here, by “single-gene” you mean mitochondrial gene or any gene (nuclear and mitochondrial)?

Response: Regrets for the confusion, here we mean both single nuclear/mitochondrial gene phylogeny. We have rephrased the sentence to increase understanding. Please see lines 409-411 on page 22 of file named ‘Manuscript’.

Comment 23. Line 389: This sentence is confusing. How can whole mitochondrial genomes be used along with concatenated PCGs? The PCGs are part of the mitogenome.

Response: Apologies for the confusion. We meant to convey concatenated PCGs from the whole mitochondrial genomes were used to construct the phylogeny. Hence, sentence rephrased in the text. Please see lines 411-414 on page 22 of file named ‘Manuscript’.

Comment 24. Line 398: I suggest rephrasing this sentence to “The mitogenome-based divergence analysis suggest that…”

Response: As advised in Comment no. 9 all ‘genome-based’ phrases were changed to ‘species-level’. Hence, the sentence was further modified to include ‘species-level’ and ‘mitogenome-based’ terminologies. Please see lines 421-425 on page 22 of file named ‘Manuscript’.

Comment 25. Lines 402-404: I did not understand what topology was presented before.

Response: It is intended to mean both the ML and BI based trees have same topologies, no previous topology presented. Hence, sentence rephrased for better understanding of the idea. Please see lines 426-427 on page 22 of file named ‘Manuscript’.

Comment 26. Lines 408-409: It would be reasonable to mention what kind of data the cited works were based on.

Response: Sentence rephrased to include the data mentioned in this cited studies is based on single mitochondrial (cob) and nuclear (rag-1) gene. Please see lines 431-433 on page 22 of file named ‘Manuscript’.

Comment 27. Lines 409-411: This indicates that the mitochondrial DNA of this species have common ancestry. This pattern does not necessarily imply a paraphyletic genus. It may be caused by population level phenomena such as introgression.

Response: Mitochondrial DNA of the species share common ancestry. Population level phenomenon such as introgression and ILS has been mentioned and discussed in detail in the ‘Major issue’ part of the rebuttal. We do acknowledge the phenomenon of introgression and ILS may cause such phylogenetic incongruences. However, the evidence from previous studies and our divergence analysis, phylogenetic analysis definitely weigh-in support of a non-monophyletic Psittacula genus. Please see lines 437-454 on page 23 of file named ‘Manuscript’.

28. Lines 415-419: All these statements are assuming that the mitogenome is always identical to the species phylogeny, which is not true. Also, this part of the text is very confusing. I was not able to understand why there are non-monophiletic clustering os 4 groups.

Response: Our study put forwards various facts and references in support of mitogenome based phylogeny to show that, more often than not it reflects true species phylogeny. The part of the manuscript mentioned in this comment has been modified to incorporate better arguments in support of our logic. Various reports have shown phylogenetic analysis associated with different genomic loci (nuclear, mitochondrial, coding, non-coding) may produce distinct phylogenetic trees respectively (Rheindt and Edwards, 2011; Stiller and Zhang, 2019). However mitogenomes have shown to provide consistent phylogenetic results in events of speciation and such consistency is even maintained in events of introgression and ILS also, wherein studies have shown levels of introgression in mitochondrial DNA to be nearly zero or significantly lesser than nuclear DNA introgression (Rheindt and Edwards, 2011; Stiller and Zhang, 2019). Such unique qualities of mitogenome eliminates stochastic errors in phylogenetic studies where the phylogenetic signal often gets masked with homoplasmy, multiple substitutions on the same site, and heterogeneous base composition (Stiller and Zhang, 2019). Such evidences support that the phylogenetic tree constructed using whole mitogenome more often than not reflects true species phylogeny. Please see lines 454-469 on page 23 of file named ‘Manuscript’. 

The clustering pattern within the Psittacula genus and arguments in support of non-monophyletic nature has been provided in the second part of the ‘Major issue’ comment, kindly consider. Please see lines 470-485 on page 23, 24 of file named ‘Manuscript’.

• Rheindt FE, Edwards SV. Genetic introgression: an integral but neglected component of speciation in birds. The Auk. 2011 Oct 1;128(4):620-32.

• Stiller J, Zhang G. Comparative phylogenomics, a stepping stone for bird biodiversity studies. Diversity. 2019 Jul;11(7):115.

29. Lines 424-433: This analysis is pointless, unless the results are discussed and compared to other birds/vertebrates. Besides that, are the dN/dS values statistically significant?

Response: The main aim of this study was to report the complete mitochondrial genome sequence of Psittacula cyanocephala and its comparison with other Psittaciformes. Hence we have not included birds of other families or other vertebrates. Regarding the evolutionary analysis, we calculated the dN/dS values of 13 protein coding genes of Psittacula cyanocephala with respect to select Psittaciformes mitogenomes and have finally reported the average value for all 13 protein coding genes. The aim was to observe whether those genes were naturally selected or under mutational pressure. Since dN/dS values of all protein coding genes were less than 1, we concluded the presence of natural selection over those genes. And the values obtained are statistically significant. Please see lines 491-499 on page 25 of file named ‘Manuscript’.

30. Lines 441-443: Change “least” to “lower”.

Response: Word changed in manuscript to ‘lower’. Please see lines 508 on page 26 of file named ‘Manuscript’.

31. Line 444: Change “multiple phyla” to “multiple clades/groups”.

Response: Rephrased in manuscript to ‘multiple clades/groups’. Please see line 510 on page 26 of file named ‘Manuscript’.

32. Line 446: Change “phylogeography” to “phylogeographic”.

Response: Word changed in manuscript. Please see line 512 on page 26 of file named ‘Manuscript’.

---

## [Decision Letter · Decision Letter 1]

16 Feb 2021

PONE-D-20-31563R1

Complete mitogenome of endemic Plum-headed parakeet Psittacula cyanocephala  – characterization and phylogenetic analysis

PLOS ONE

Dear Dr. SINGH,

Thank you for submitting your manuscript to PLOS ONE. After careful consideration, we feel that it has merit but does not fully meet PLOS ONE’s publication criteria as it currently stands. Therefore, we invite you to submit a revised version of the manuscript that addresses the points raised during the review process.

We look forward to receiving your revised manuscript.

Kind regards,

Maria Andreína Pacheco, Ph.D.

Academic Editor

PLOS ONE

Additional Editor Comments (if provided):

Please, pay attention to the reviewer’s comment and incorporate then in a new version.

Reviewers' comments:

Reviewer's Responses to Questions

**Comments to the Author**

1. If the authors have adequately addressed your comments raised in a previous round of review and you feel that this manuscript is now acceptable for publication, you may indicate that here to bypass the “Comments to the Author” section, enter your conflict of interest statement in the “Confidential to Editor” section, and submit your "Accept" recommendation.

Reviewer #2: (No Response)

Reviewer #3: All comments have been addressed

2. Is the manuscript technically sound, and do the data support the conclusions?

Reviewer #2: Yes

Reviewer #3: Partly

3. Has the statistical analysis been performed appropriately and rigorously? 

Reviewer #2: Yes

Reviewer #3: Yes

4. Have the authors made all data underlying the findings in their manuscript fully available?

Reviewer #2: No

Reviewer #3: Yes

5. Is the manuscript presented in an intelligible fashion and written in standard English?

Reviewer #2: Yes

Reviewer #3: Yes

6. Review Comments to the Author

Reviewer #2: The authors have incorporated most suggestions and I can say that they addressed my main concerns in the last review satisfactorily. However, I have a few minor comments that the authors may want to consider.

- Lines 197-198: “Evolutionary constraints… were…”

- Line 205: “The codons used … and were …”

- Lines 197-207: to my knowledge, PAL2NAL is not a software from PAML package, but CODEML is. The authors should revise this.

- Lines 234-235: I insist that the authors should report at least one statistic that they used to check the convergence of the Bayesian chains (average standard deviation of the split frequencies…) and one statistic to check the mixing of the Bayesian chains (ESS…).

- Line 415: “The evolutionary rate…”

- The alignment data as well as the phylogenetic trees (nwk/nexus) should be made available (at the supporting material or a public repository)

Reviewer #3: The autors did a really good job improving the manuscript by addressing all the previous reviewers’ concerns. The manuscript is mainly focused on the building and characterization of the genome of Psittacula cyanocephala, and this is the main strenght of it. Phylogenetic analyses, however, as mentioned by one of the previous reviewers, did not provide any new insights on Psittacula parakeets phylogenetic relationships, even thought show support to previous claims on Psittacula paraphyly. The sampling scheme of the includes a limited number of taxa because of the few available mitogenomes (6 of the 16 Psittacula species and only 1 of the 4 Tanygnathus species) and because of this, it only shows the potential applications of the use of mitogenomes for phylogenetic analyses. Phylogenetic analyses also include data from 44 parrot species but these do not provide new information on the relationships within Psittaciformes.

7. PLOS authors have the option to publish the peer review history of their article (what does this mean?). If published, this will include your full peer review and any attached files.

Reviewer #2: No

Reviewer #3: No

---

## [Author Response · Author response to Decision Letter 1]

25 Feb 2021

Ms. Ref. No.: PONE-D-20-31563R1 - EMID: 75cc66b576b95812

Title: Complete mitogenome of endemic Plum-headed parakeet Psittacula cyanocephala – characterization and phylogenetic analysis

Response to reviewers

We would like to thank Dr. Maria Andreína Pacheco (Academic Editor) and the Reviewers for their constructive remarks. We have revised the manuscript as suggested by the Reviewers. The revised manuscript is submitted for further consideration. Please find below a point-by-point rebuttal to the issues raised.

Additional Editor Comments:

Comment 1: Please, pay attention to the reviewer’s comment and incorporate then in a new version. 

Response: We have carefully addressed the reviewer’s comment in the new version of the manuscript. 

Reviewer #2

Comment 1. Lines 197-198: “Evolutionary constraints… were…” 

Response: Sentence rephrased in the manuscript. Please see lines 197-198 on page 10 of file named ‘Manuscript’.

Comment 2. Line 205: “The codons used … and were …”

Response: Sentence rephrased in the manuscript. Please see lines 205 on page 10 of file named ‘Manuscript’.

Comment 3. Lines 197-207: to my knowledge, PAL2NAL is not a software from PAML package, but CODEML is. The authors should revise this.

Response: PAL2NAL is a programme embedded in PAML. PAL2NAL automatically calculates dS and dN using codeml program in PAML (http://www.bork.embl.de/pal2nal/#Ref).

Comment 4. Lines 234-235: I insist that the authors should report at least one statistic that they used to check the convergence of the Bayesian chains (average standard deviation of the split frequencies…) and one statistic to check the mixing of the Bayesian chains (ESS…). 

Response: Thanks for the suggestion. To assess convergence, the average standard deviation of split frequencies was calculated. The effective sample size value of the trace was also diagnosed to confirm the mixing of the Bayesian Markov chains. Please see lines 236-239 on page 11, 12 of file named ‘Manuscript’. The resulting values and data are submitted as additional supplementary information.

Comment 5. Line 415: “The evolutionary rate…”

Response: Sentence rephrased in the manuscript. Please see line 410 on page 22 of file named ‘Manuscript’.

Comment 6. The alignment data as well as the phylogenetic trees (nwk/nexus) should be made available (at the supporting material or a public repository).

Response: The alignment file, both BI and ML tree files are submitted as additional supplementary information. 

Reviewer #3

Comment 1. The authors did a really good job improving the manuscript by addressing all the previous reviewers’ concerns. The manuscript is mainly focused on the building and characterization of the genome of Psittacula cyanocephala, and this is the main strength of it. Phylogenetic analyses, however, as mentioned by one of the previous reviewers, did not provide any new insights on Psittacula parakeets phylogenetic relationships, even thought show support to previous claims on Psittacula paraphyly. The sampling scheme of the includes a limited number of taxa because of the few available mitogenomes (6 of the 16 Psittacula species and only 1 of the 4 Tanygnathus species) and because of this, it only shows the potential applications of the use of mitogenomes for phylogenetic analyses. Phylogenetic analyses also include data from 44 parrot species but these do not provide new information on the relationships within Psittaciformes.

Response: Thank you very much for your thoughtful comments and insight. The complete mitogenome of Psittacula cyanocephala is necessary to build the complete phylogeny of Psittacula genus complex. Our work does lend support to paraphyly within Psittacula genus as indicated by previous studies. Evidences from multiple sources (complete mitogenomes, single nuclear/mitochondrial genes etc.) are necessary to propose and validate a comprehensive phylogeny (Tietze DT, 2018). We believe our study is an important contribution in this aspect. 

• Tietze DT. Bird species: how they arise, modify and vanish. Springer Nature; 2018.

---

## [Editor Report · Decision Letter 2]

2 Mar 2021

Complete mitogenome of endemic Plum-headed parakeet Psittacula cyanocephala  – characterization and phylogenetic analysis

PONE-D-20-31563R2

Dear Dr. SINGH,

We’re pleased to inform you that your manuscript has been judged scientifically suitable for publication and will be formally accepted for publication once it meets all outstanding technical requirements.

Kind regards,

Maria Andreína Pacheco, Ph.D.

Academic Editor

PLOS ONE

---

## [Editor Report · Acceptance letter]

23 Mar 2021

PONE-D-20-31563R2 

**Complete mitogenome of endemic Plum-headed parakeet *Psittacula cyanocephala* – characterization and phylogenetic analysis**

Dear Dr. Singh:

I'm pleased to inform you that your manuscript has been deemed suitable for publication in PLOS ONE. Congratulations! Your manuscript is now with our production department. 

Kind regards, 

on behalf of

Dr. Maria Andreína Pacheco 

Academic Editor

PLOS ONE